# Na+ influx *via* Orai1 inhibits intracellular ATP-induced mTORC2 signaling to disrupt CD4 T cell gene expression and differentiation

Yong Miao[1], Jaya Bhushan[1], Adish Dani[1,2], Monika Vig[1]*

[1]Department of Pathology and Immunology, Washington University School of Medicine, St Louis, United States; [2]Hope Center for Neurological Disorders, Washington University School of Medicine, St Louis, United States

**Abstract** T cell effector functions require sustained calcium influx. However, the signaling and phenotypic consequences of non-specific sodium permeation *via* calcium channels remain unknown. $\alpha$-SNAP is a crucial component of Orai1 channels, and its depletion disrupts the functional assembly of Orai1 multimers. Here we show that $\alpha$-SNAP hypomorph, hydrocephalus with hopping gait, *Napa$^{hyh/hyh}$* mice harbor significant defects in CD4 T cell gene expression and Foxp3 regulatory T cell (Treg) differentiation. Mechanistically, TCR stimulation induced rapid sodium influx in *Napa$^{hyh/hyh}$* CD4 T cells, which reduced intracellular ATP, $[ATP]_i$. Depletion of $[ATP]_i$ inhibited mTORC2 dependent NF$\kappa$B activation in *Napa$^{hyh/hyh}$* cells but ablation of Orai1 restored it. Remarkably, TCR stimulation in the presence of monensin phenocopied the defects in *Napa$^{hyh/hyh}$* signaling and Treg differentiation, but not IL-2 expression. Thus, non-specific sodium influx *via* bonafide calcium channels disrupts unexpected signaling nodes and may provide mechanistic insights into some divergent phenotypes associated with Orai1 function.

**\*For correspondence:** mvig@ WUSTL.EDU

**Competing interests:** The authors declare that no competing interests exist.

## Introduction

A sustained rise in cytosolic calcium is necessary for nuclear translocation of calcium-dependent transcription factors such as nuclear factor of activated T cell (NFAT) (*Crabtree, 1999*; *Parekh and Putney, 2005*; *Winslow et al., 2003*; *Macian, 2005*; *Vig and Kinet, 2009*, *2007*; *Crabtree, 2001*). NFAT proteins are essential for the development of several tissues but have been found to be dispensable for thymic development and function of Foxp3 regulatory T cells (Tregs) (*Crabtree and Olson, 2002*; *Crabtree, 2001*, *1999*; *Timmerman et al., 1997*; *Vaeth et al., 2012*). The role of proteins directly involved in sustained calcium influx, however, remains less well established. Specifically, genetic ablation of ORAI1, the pore forming subunit of calcium release activated calcium (CRAC) channels (*Vig et al., 2006b*, *2006a*; *Peinelt et al., 2006*), partially inhibits T cell effector cytokines in mice and does not affect Foxp3 Treg development (*Vig et al., 2008*; *Vig and Kinet, 2009*; *Gwack et al., 2008*; *McCarl et al., 2010*). The role of ORAI2 as well as ORAI3, the two closely related homologs of ORAI1 that are highly expressed in mouse T cells remains unestablished in mice and humans although all ORAIs are capable of reconstituting CRAC currents in vitro (*Mercer et al., 2006*; *Lis et al., 2007*; *DeHaven et al., 2007*).

STIM1 and STIM2, the ER resident calcium sensor proteins, are required for ER calcium release, Orai1 activation and T lymphocyte effector functions (*Oh-Hora et al., 2008*). However, ablation of STIMs, but not ORAIs, affects thymic development of Tregs (*Oh-Hora et al., 2013*; *McCarl et al., 2010*) and ablation of STIMs, but not ORAI1, results in multi-organ autoimmunity in mice and

humans (*Oh-Hora et al., 2008*; *McCarl et al., 2010*; *Picard et al., 2009*). Because STIMs perform several additional functions such as regulation of calcium selectivity of ORAI1 channels (*McnallyMcNally et al., 2012*) as well as inhibition of voltage-gated calcium channel Cav1.2 (*Wang et al., 2010*; *Park et al., 2010*), role of sustained calcium influx or store-operated calcium entry (SOCE) in the development of Tregs and autoimmunity remains correlative (*Oh-Hora et al., 2008*, *2013*). Likewise, the phenotypes of human patients harboring different Stim and Orai mutations range from immunodeficiency to autoimmunity and cancer. Despite this diversity, all phenotypes are currently correlated with reduced SOCE (*Picard et al., 2009*).

We have previously shown that α-soluble NSF-attachment protein (α-SNAP), a cytosolic protein traditionally studied in the context of soluble NSF attachment protein receptor (SNARE) complex disassembly and membrane trafficking (*Clary et al., 1990*), directly binds Stim1 and Orai1 and is necessary for the functional assembly and ion specificity of multimeric Orai1 channels (*Miao et al., 2013*; *Li et al., 2016*). In addition, α-SNAP has been implicated in AMP kinase (AMPK) inhibition and zippering of SNAREs in vitro (*Park et al., 2014*; *Baur et al., 2007*; *Wang and Brautigan, 2013*). SNAREs play a direct role in exocytosis and are therefore required for cytotoxic T, natural killer and mast cell degranulation (*Baram D et al., 2001*; *Puri et al., 2003*; *Hepp et al., 2005*; *Suzuki and Verma, 2008*). However, the role of α-SNAP is less clear in vivo, and remains unexplored in the immune system. α-SNAP deletion is embryonic lethal in mice and a hypomorphic missense mutation in α-SNAP, hydrocephalus with hopping gait, (*Napa*[hyh/hyh]) has been previously reported to cause neuro-developmental defects (*Bronson and Lane, 1990*; *Chae et al., 2004*; *Hong et al., 2004*).

Here, we show that reduced expression of α-SNAP causes unexpected defects in CD4 T cell signaling, gene expression and Foxp3 Treg differentiation. Using RNAi-mediated ablation of Orai1 in *Napa*[hyh/hyh] CD4 T cells and monensin treatment of wildtype CD4 T cells, we demonstrate that Orai1 mediated sodium influx, but not reduced SOCE, depletes [ATP]$_i$ in T cell receptor (TCR) stimulated *Napa*[hyh/hyh] CD4 T cells. Furthermore, we find that depletion of [ATP]$_i$ levels disrupts mTORC2 activation which, in turn, inhibits NFκB activation and differentiation of Foxp3 Tregs in *Napa*[hyh/hyh] mice in vivo as well as in vitro. Therefore, analysis of α-SNAP deficient mice reveals that sodium permeation *via* Orai1 disrupts a novel signaling node and could provide alternate mechanistic insights into the diversity of phenotypes observed in Stim and Orai mutant human patients.

## Results

### *Napa*[hyh/hyh] mice harbor severe defects in the production of CD4 T cell effector cytokines

Mice bearing *Napa*[hyh/hyh] mutation on a mixed background have been characterized previously in the context of neurodevelopmental disorders (*Bronson and Lane, 1990*; *Chae et al., 2004*; *Hong et al., 2004*). We backcrossed *Napa*[hyh/hyh] mice on to C57BL/6 background and found that homozygous mutant *Napa*[hyh/hyh] mice were significantly smaller in size and died perinatally, within 2–3 weeks. To overcome the issue of perinatal lethality, we generated fetal liver chimeras using irradiated CD45.1+ congenic recipients reconstituted with CD45.2+ wildtype or *Napa*[hyh/hyh] E15.5 embryos. We analyzed fetal liver chimeras at 8–12 week post-transfer and found that the reconstitution efficiency and number of thymocytes (*Figure 1A*) and splenocytes (*Figure 1B*) was comparable in wildtype (WT) and *Napa*[hyh/hyh] chimeras. Relative abundance of CD4 and CD8 T cells in the thymus (*Figure 1C*) and spleen (*Figure 1D*) was also normal in *Napa*[hyh/hyh] fetal liver chimeras. Therefore, we performed all the subsequent analysis of wildtype and *Napa*[hyh/hyh] CD4 T cells and Foxp3 Tregs using fetal liver chimeras, unless otherwise specified.

α-SNAP null mice are embryonic lethal and, in accordance with previous reports, *Napa*[hyh/hyh] CD4 T cells showed ~40% depletion of α-SNAP levels (*Figure 1E*). Given the role of α-SNAP in SNARE recycling (*Clary et al., 1990*), we first compared the levels of cell surface receptors. Surprisingly, surface expression TCR and co-receptors was normal in *Napa*[hyh/hyh] peripheral CD4 T cells (*Figure 1F*). Resting *Napa*[hyh/hyh] T lymphocytes showed largely normal surface expression of CD25, CD44 and CD69 and their up-regulation following receptor-mediated stimulation was comparable to WT (*Figure 1G*).

CRAC channel components, Orai1 and Stim1 are necessary for optimal production and secretion of several T cell effector cytokines (*Vig et al., 2008*; *Vig and Kinet, 2009*; *Gwack et al., 2008*; *Oh-*

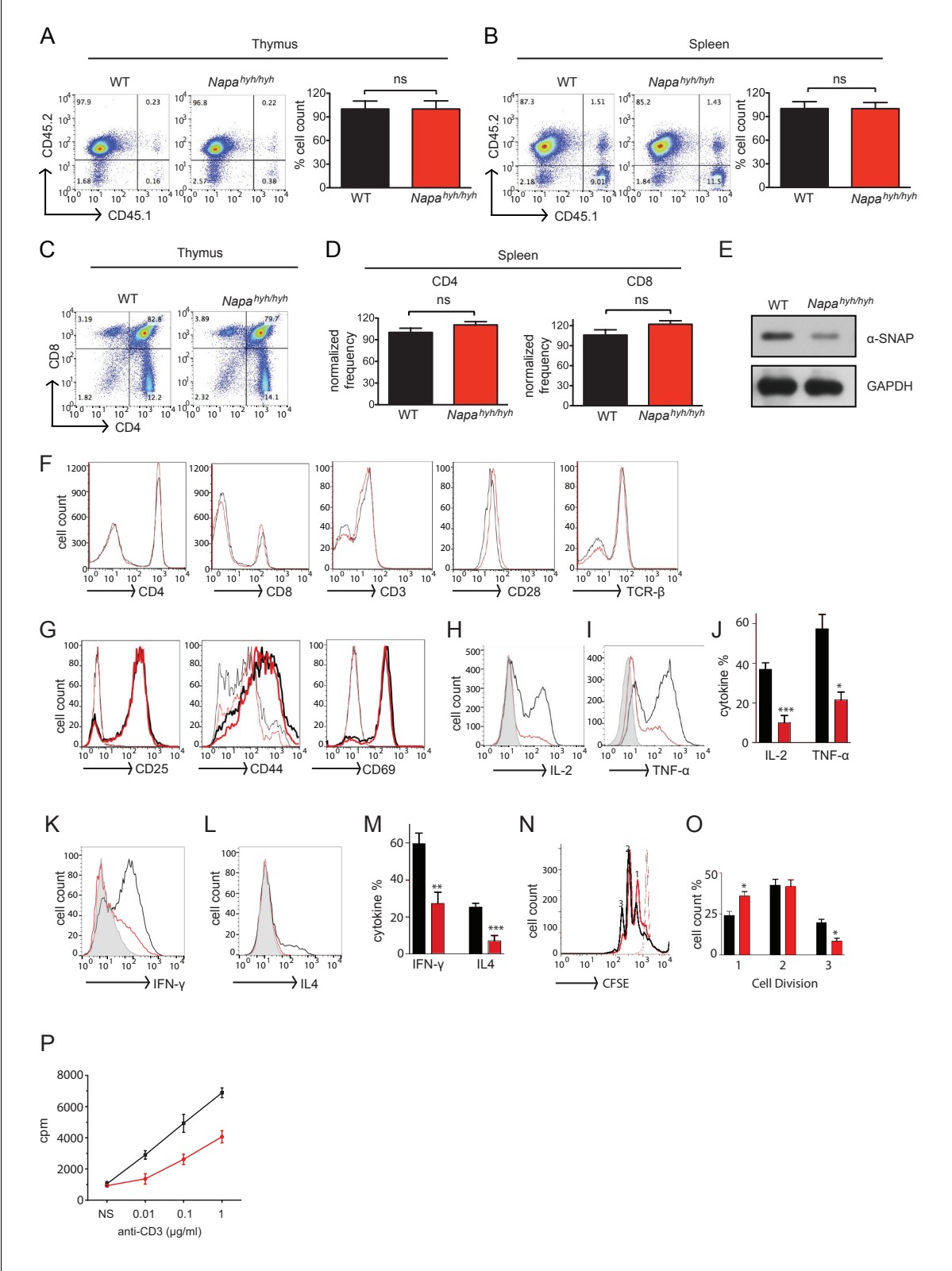

**Figure 1.** *Napa^hyh/hyh* mice harbor severe defects in the production of CD4 T cell effector cytokines. (A and B) Representative FACS profile showing the reconstitution efficiency and average cell yields from the thymus (A) and spleen (B) of WT (black) and *Napa^hyh/hyh* (red) fetal liver chimeric mice. (n = 25). (C and D) Representative FACS profile showing the percentage of CD4+, CD8+ single and double positive thymocytes in CD45.2+ gated cells from WT and *Napa^hyh/hyh* chimera thymus (C) and spleen (D). (n = 10). (E) Representative Western blot for α-SNAP in whole cell lysates prepared from WT and

*Figure 1 continued on next page*

*Figure 1 continued*

*Napa*^*hyh/hyh* lymph node cells. (n > 5). (**F**) FACS profiles showing surface staining of WT (black) and *Napa*^*hyh/hyh* (red) spleen cells with anti-CD4, anti-CD8, anti-CD3, anti-CD28 and anti-TCRβ antibodies respectively. (n = 5). (**G**) FACS profiles of resting (thin lines) and receptor stimulated (thick lines); WT (black) and *Napa*^*hyh/hyh* (red) CD4 T cells stained for various activation markers. (n = 3). (**H–J**) FACS profiles showing intracellular staining for IL-2 (H, J) and TNF-α (I,J) in WT (black) and *Napa*^*hyh/hyh* (red) CD4 T cells 6 hr post-stimulation. Grey peak shows unstimulated control. (n = 5 repeats from five chimeras each). (**K–M**) FACS profiles showing intracellular cytokine staining for Th1 (K,M) and Th2 (L,M) signature cytokines in polarized WT (black) and *Napa*^*hyh/hyh* (red) CD4 T lymphocytes. Grey peak shows undifferentiated control. (n = 3). (**N,O**) CFSE dilution profiles (N) and their quantifications (O), showing proliferation of WT (black) and *Napa*^*hyh/hyh* (red) CD4 T cells in response to plate bound anti-CD3 and anti-CD28 antibodies. Light traces show unstimulated control. (n = 3). (**P**) Representative plot showing proliferation of WT (black) and *Napa*^*hyh/hyh* (red) splenocytes in response to soluble anti-CD3 and anti-CD28, estimated using $^3$H thymidine incorporation. (n = 3). (See also *Figure 1—figure supplement 1*).

The following figure supplement is available for figure 1:

**Figure supplement 1.** Bar plots showing the average MFIs of the intracellular staining for T-bet and GATA-3 in Th1 and Th2 differentiated WT (black) and *Napa*^*hyh/hyh* (red) CD4 T cells, respectively. (n = 3).

*Hora et al., 2008*). However, given a partial depletion of α-SNAP in *Napa*^*hyh/hyh* mice, we first sought to determine whether *Napa*^*hyh/hyh* CD4 T cells showed defects in the production of effector cytokines. Surprisingly, we found significant defects in IL-2 (*Figure 1H and J*) and TNF-α production by TCR-stimulated *Napa*^*hyh/hyh* CD4 T cells (*Figure 1I and J*). *Napa*^*hyh/hyh* CD4 T cells cultured under T helper 1 (Th1) polarizing conditions showed a minor defect in IFN-γ production (*Figure 1K and M*), however, we observed a robust defect in IL-4 expression in Th2-polarized *Napa*^*hyh/hyh* CD4 T cells (*Figure 1L and M*). Intracellular levels of T-bet or Gata-3 did not appear to be significantly altered in *Napa*^*hyh/hyh* mice (*Figure 1—figure supplement 1*). Furthermore, *Napa*^*hyh/hyh* CD4 T cells (*Figure 1N–1O*) and splenocytes (*Figure 1P*) showed a partial defect in anti-CD3-induced proliferation. Taken together, these data demonstrate that *Napa*^*hyh/hyh* CD4 T lymphocytes harbor a significant defect in the production of several key effector cytokines, while exhibiting normal levels of cell surface receptors.

## *Napa*^*hyh/hyh* mice harbor significant defects in the differentiation of Foxp3 regulatory T cells in vivo and in vitro

*Stim1-/-Stim2-/-* mice (*Oh-Hora et al., 2013*), but neither *Orai1-/-* (*McCarl et al., 2010*) nor *Nfatc1-/-Nfatc2-/-* mice (*Vaeth et al., 2012*), harbor defects in the development of thymic Foxp3 Tregs. Interestingly, analysis of *Napa*^*hyh/hyh* fetal liver chimeras of 8–12 weeks showed lower percentage (*Figure 2A*) as well as total number (*Figure 2—source data 1*) of thymic Foxp3 Tregs when compared to WT. Mixed fetal liver chimeras of WT and *Napa*^*hyh/hyh* showed further reduced percentages of *Napa*^*hyh/hyh* Foxp3 Tregs in the thymus (*Figure 2B*) as well as peripheral lymphoid tissues (*Figure 2C*) and *Napa*^*hyh/hyh* Foxp3 Tregs consistently showed lower surface expression of CD44 (*Figure 2D*) and GITR (*Figure 2E*). These data suggest additional potential defects in the homing (*Luo et al., 2016*) and in vivo expansion of Foxp3 Tregs (*Ronchetti et al., 2015*; *Ephrem et al., 2013*; *Liao et al., 2010*). Indeed, lamina propria of *Napa*^*hyh/hyh* chimeras showed further reduced numbers of Foxp3 Tregs (*Figure 2F*). Furthermore, in vitro differentiation of *Napa*^*hyh/hyh* CD4 T cells also yielded lower percentage (*Figure 2G*) and number (*Figure 2—source data 2*) of Foxp3 iTregs. Taken together, these data demonstrate a crucial role for α-SNAP in Foxp3+ Treg differentiation in vivo as well as in vitro.

## Orai1-mediated sodium influx inhibits Foxp3 iTreg differentiation by disrupting NFκB activation in *Napa*^*hyh/hyh* CD4 T cells

We have previously shown that α-SNAP is an integral component of the CRAC channel complex, where it facilitates the functional assembly as well as ion selectivity of Orai1 multimers via specific molecular interactions (*Li et al., 2016*). Because *Orai1-/-* mice do not show a defect in Foxp3 regulatory T cell development (*Vig et al., 2008*; *Vig and Kinet, 2009*; *Gwack et al., 2008*; *McCarl et al., 2010*), we hypothesized that non-specific sodium permeation *via* Orai1 could lead to reduced generation of thymic and iTregs in *Napa*^*hyh/hyh* mice.

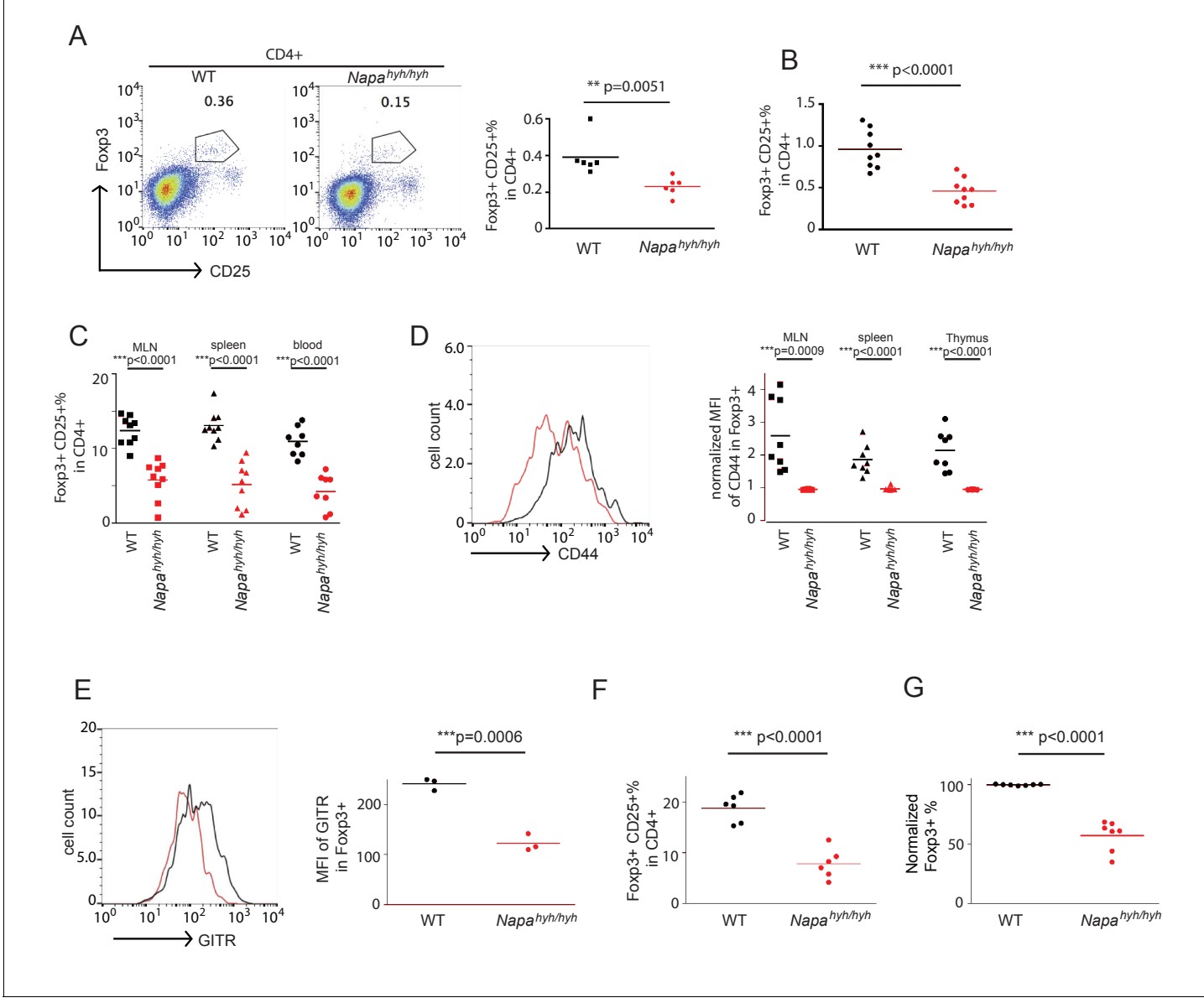

**Figure 2.** *Napa^hyh/hyh* mice harbor significant defects in the differentiation of Foxp3 regulatory T cells in vivo and in vitro. (**A**) Representative FACS profiles and dot plot showing the percentage of Foxp3+CD25+ cells in the CD4+CD45.2+ population from WT and *Napa^hyh/hyh* chimera thymii. (n = 3 with 2 chimeras/experiment). (**B**) Dot plot showing the percentage of WT and *Napa^hyh/hyh* Foxp3+ cells in mixed chimera thymii. (n = 4 with 2–3 chimeras/experiment). *p* value from paired student's t-test. (**C**) Dot plots showing the percentage of WT and *Napa^hyh/hyh* Foxp3+ cells in blood, spleen and mesenteric lymph nodes (MLN). (n = 4 with 2–3 chimeras/experiment). (**D**) Representative FACS profile and dot plots showing mean fluorescence intensity (MFI) of surface expression of CD44 on WT (black) and *Napa^hyh/hyh* (red) Foxp3+ cells from various lymphoid tissues of mixed chimeras. (n = 4 with 2 chimeras/experiment). (**E**) FACS profile and dot plot showing mean fluorescence intensity (MFI) of the surface expression of GITR on WT (black) and *Napa^hyh/hyh* (red) Foxp3+ cells from spleen. (n = 3). (**F**) Dot plot showing the percentage of WT and *Napa^hyh/hyh* Foxp3+ cells in the lamina propria CD4 T cells isolated from mixed chimeras. (n = 3 with 2 chimeras/experiment). (**G**) Dot plot showing the normalized percentage of WT (black) and *Napa^hyh/hyh* (red) Foxp3+ cells in in vitro-differentiated CD4 T cells. (n = 7). *p* value from paired student's t test. (See also *Figure 2—source data 1* and *2*).

The following source data is available for figure 2:

**Source data 1.** Thymic Foxp3 Tregs in WT and *Napa^hyh/hyh* chimeras.
**Source data 2.** Foxp3 iTregs in WT and *Napa^hyh/hyh* CD4 T cell cultures.

In agreement with our previous findings (*Miao et al., 2013*), stimulation of *Napa^hyh/hyh* CD4 T cells *via* TCR (*Figure 3A and B*) or Thapsigargin (TG) (*Figure 3C*) induced lower SOCE. More strikingly though, *Napa^hyh/hyh*, but not wildtype CD4 T cells, showed rapid and significant sodium entry in response to TCR as well as TG stimulation (*Figure 3D and E*). Interestingly, RNAi-mediated depletion of Orai1 in *Napa^hyh/hyh* cells abolished sodium influx, demonstrating that sodium enters *via* Orai1 in TCR-stimulated *Napa^hyh/hyh* CD4 T cells (*Figure 3F*). Furthermore, replacement of extracellular sodium with a membrane impermeable organic monovalent cation, N-methyl-D-glucamine (NMDG) prevented fluorescence shift of the sodium dye, SBFI (*Figure 3F*), establishing its specificity for sodium and the direction of sodium flux in receptor-stimulated CD4 T cells. Of note, treatment of wildtype CD4 T cells with monensin, a non-specific sodium ionophore, induced similar levels of sodium influx (*Figure 3G*) when compared to TCR-stimulated *Napa^hyh/hyh* CD4 T cells.

In agreement with reduced SOCE, we observed that nuclear translocation of NFAT was defective in *Napa^hyh/hyh* CD4 T cells (*Figure 3H*). However, nuclear translocation of NFκB p65 and c-Rel transcription factors was also severely inhibited in TCR-stimulated *Napa^hyh/hyh* CD4 T cells (*Figure 3H and I*), although we found no significant defect in T cell receptor-proximal signaling events or MAPK activation (*Figure 3J and K*). TCR proximal signaling requires an interplay of several cell surface receptors, co-receptors and membrane proximal kinases, thus further reinforcing our observations that membrane receptor signaling events remain unperturbed in *Napa^hyh/hyh* T cells.

To determine whether defects in NFκB translocation were due to reduced SOCE or non-selective sodium influx, we first depleted Orai1 expression in CD4 T cells using RNAi. Orai1 depletion lead to reduced SOCE (*Figure 3L*) and nuclear translocation of NFAT (*Figure 3M*). However, NFκB activation (*Figure 3N*) and iTreg differentiation (*Figure 3O*) were normal in Orai1-depleted CD4 T cells. On the other hand, stimulation of wildtype CD4 T cells in the presence of monensin did not affect IL-2 expression (*Figure 3P*) or NFAT activation (*Figure 3Q*), but inhibited NFκB activation (*Figure 3R*) and iTreg differentiation (*Figure 3S*). Taken together, these data show that TCR-induced non-selective sodium influx *via* Orai1 inhibits NFκB activation to restrict Foxp3 T cell development in *Napa^hyh/hyh* mice.

## TCR-induced non-specific sodium influx depletes [ATP]$_i$ in *Napa^hyh/hyh* CD4 T cells

Next, by examining additional signaling events in TCR-stimulated *Napa^hyh/hyh* CD4 T cells, we sought to understand the molecular basis by which NFκB activation is defective. The sodium potassium ATPase (Na K ATPase) maintains resting membrane potential by pumping out intracellular sodium using ATP hydrolysis and can consume ~30–40% of cellular ATP at any given time in resting cells (*Torres-Flores et al., 2011*). Hence, we reasoned that TCR-induced abnormal sodium entry could cause an acute metabolic burden in *Napa^hyh/hyh* cells and measured the change in [ATP]$_i$ levels in resting and TCR-stimulated CD4 T cells at different time points post stimulation. Interestingly, wildtype CD4 T cells showed a rapid and significant rise in [ATP]$_i$ upon TCR ligation (*Figure 4A*). *Napa^hyh/hyh* CD4 T cells failed to show this increase but instead showed a sustained drop in [ATP]$_i$ post-stimulation (*Figure 4A*).

To determine whether the defect in [ATP]$_i$ rise was due to sodium influx, we exchanged extracellular sodium with NMDG, and found that the [ATP]$_i$ rise was largely restored in *Napa^hyh/hyh* T cells (*Figure 4A*). Stimulation of wildtype CD4 T cells in the presence of monensin also depleted [ATP]$_i$ levels (*Figure 4B*) and RNAi-mediated depletion of Orai1 in *Napa^hyh/hyh* CD4 T cells largely restored the [ATP]$_i$ depletion (*Figure 4C*).

Naïve CD4 T cells depend on oxidative phosphorylation to generate [ATP]$_i$. To determine whether the mitochondrial number and function were normal in *Napa^hyh/hyh* CD4 T cells, we stained *Napa^hyh/hyh* CD4 T cells with MitoTracker green and found that their mitochondrial content was normal (*Figure 4D*). Oxygen consumption rate (OCR), extracellular acidification rate (ECAR) and the respiratory capacities of naïve (*Figure 4E and F*), receptor-stimulated (*Figure 4G and H*) and TH0-differentiated (*Figure 4I and J*) *Napa^hyh/hyh* CD4 T cells were also largely comparable to wildtype T cells. Taken together, these data demonstrate that non-specific sodium influx, but not compromised mitochondrial health, underlies depletion of [ATP]$_i$ in receptor-stimulated *Napa^hyh/hyh* CD4 T cells.

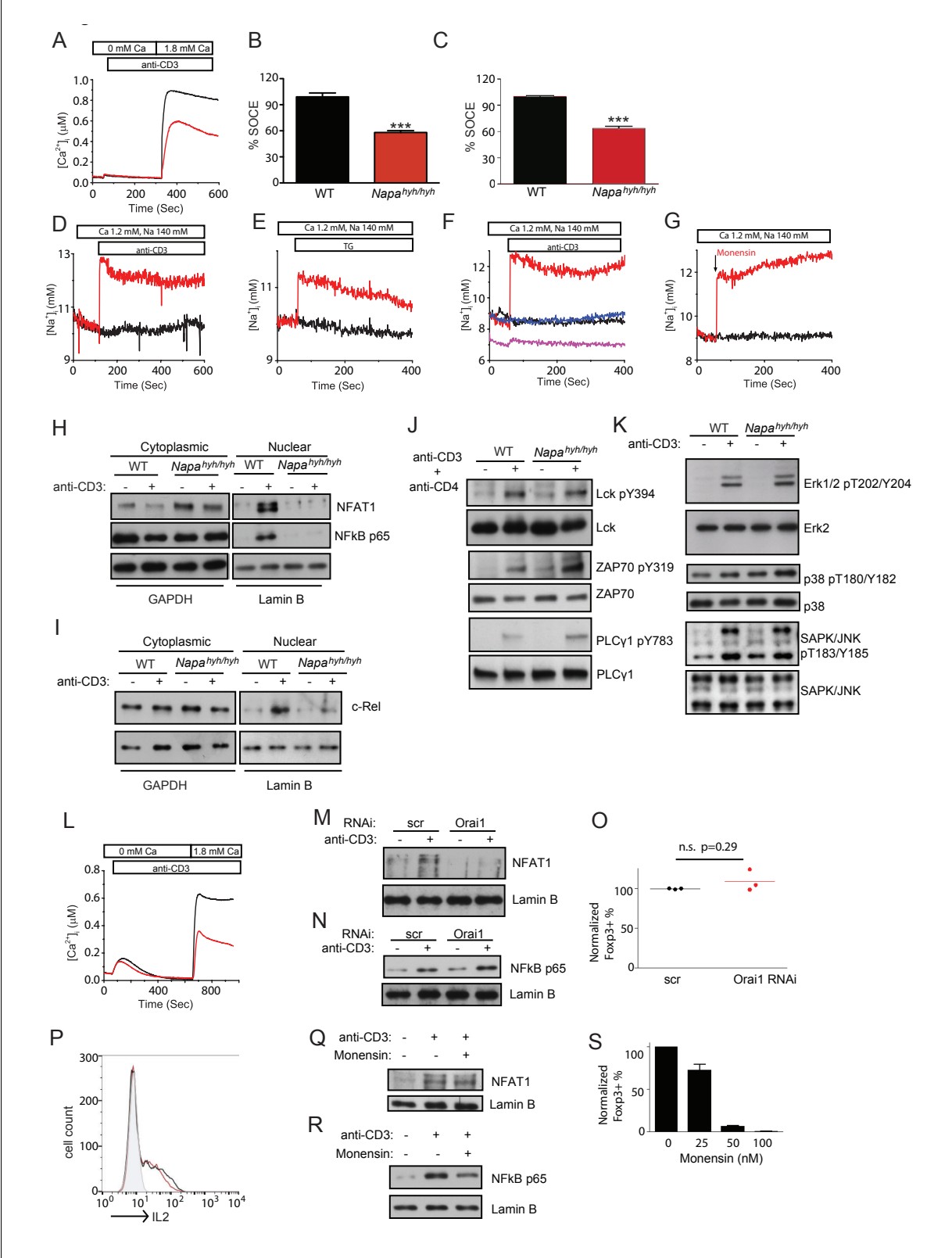

**Figure 3.** Orai1-mediated sodium influx inhibits Foxp3 iTreg differentiation by disrupting NFκB activation in *Napa^hyh/hyh* CD4 T cells. (**A–C**) Representative Fura-2 profiles showing real-time change in average cytosolic calcium levels in WT (black) or *Napa^hyh/hyh* (red) CD4 T cells stimulated with anti-CD3 antibody (**A** and **B**) (n = 5 with~50 to 100 cells per experiment) or thapsigargin (TG) (**C**) (n = 3 with~50 to 100 cells per experiment). Percent SOCE was calculated by normalizing average WT response to 100 and then calculating the % response of *Napa^hyh/hyh* CD4 T cells. (**D** and **E**)

*Figure 3 continued on next page*

Figure 3 continued

Average SBFI profiles showing real-time change in [Na]$_i$ of WT (black) and *Napa*$^{hyh/hyh}$ (red) CD4 T cells stimulated with anti-CD3 antibody (D) (n = 5 with~50 to 100 cells per experiment) or TG (E) (n = 1 with~50 to 100 cells per experiment). (F) Average SBFI profiles of anti-CD3-stimulated WT (black) and *Napa*$^{hyh/hyh}$ CD4 T cells treated with scramble (scr) RNAi (red) or Orai1 RNAi (blue); (magenta) anti-CD3-stimulated *Napa*$^{hyh/hyh}$ T cells where [Na]$_e$ was replaced with NMDG. (n = 1 with~50 to 100 cells per experiment). (G) SBFI profiles of WT CD4 T cells, treated with Monensin (red) or untreated (black). (n = 1 with~50 to 100 cells per experiment). (H,I) Western blot for cytosolic and nuclear NFAT1 and NFκB p65 (H) or c-Rel (I) in receptor-stimulated WT and *Napa*$^{hyh/hyh}$ CD4 T cells. (n = 4). (J,K) Western blot for total and phospho-Lck, ZAP-70 and PLC-γ (J) and total and phospho-Erk1/2, p38 and Jnk (K) in receptor-stimulated WT and *Napa*$^{hyh/hyh}$ CD4 T cell WCLs. (n = 3). (L) Representative Fura-2 profiles showing real-time change in average cytosolic calcium levels in scr (black) or Orai1 RNAi treated (red) CD4 T cells stimulated with anti-CD3 antibody. (n = 2 with~50 to 100 cells per experiment). (M,N) Western blot for cytosolic and nuclear NFAT (M) and NFκB p65 (N) in receptor-stimulated scr or Orai1 RNAi (Orai1) treated WT CD4 T cells. (O) Representative dot plot showing the normalized percentage of Foxp3+ cells in scr (black) and Orai1 RNAi treated (red) CD4 T cells differentiated in vitro. (n = 3), p value from paired student's t-test. (P) FACS profiles showing intracellular IL-2 expression in WT CD4 T cells, receptor-stimulated in the presence (red) or absence (black) of monensin. (n = 2). (Q,R) Western blot for nuclear NFAT1 (Q) and NFκB p65 (R) in receptor stimulated WT CD4 T cells with or without monensin. (n = 2). (S) Bar plot showing % Foxp3+ cells differentiated in vitro, in the absence or presence of different doses of monensin. (n = 3).

## Depletion of [ATP]$_i$ inhibits mTORC2 activation in *Napa*$^{hyh/hyh}$ CD4 T cells

Extracellular ATP [ATP]$_e$ has been extensively studied in the context of T cell activation (*Schenk et al., 2008*; *Ledderose et al., 2014*) autoimmunity and graft versus host disease (*Atarashi et al., 2008*; *Wilhelm et al., 2010*) but the physiological significance of TCR- induced acute rise in [ATP]$_i$ remains unknown. We hypothesized that although dispensable for TCR proximal signaling (*Figure 3J and K*), TCR-induced [ATP]$_i$ rise could be necessary to support the relatively distal signaling events following TCR activation. Recently, mTORC2 has emerged as a crucial, but complex, player in T cell differentiation (*Chi, 2012*). mTORC2 can sense a variety of upstream signals and according to one report, directly senses ATP and phosphorylates AKT Thr450 in vitro (*Chen et al., 2013*). The upstream activator of mTORC2 in CD4 T cells, however, remains unestablished (*Masui et al., 2014*; *Navarro and Cantrell, 2014*). Therefore, we assessed the phosphorylation of AKT Ser473, a well-established mTORC2 target in TCR-stimulated *Napa*$^{hyh/hyh}$ CD4 T cells. *Napa*$^{hyh/hyh}$ CD4 T cells showed significantly reduced phosphorylation of AKT Ser473 (*Figure 5A*), but not Thr308 (*Figure 5B*), the mTORC1 target site. Furthermore, TCR stimulation of wildtype CD4 T cells in the presence of monensin also blocked AKT Ser473 phosphorylation (*Figure 5C*) but not Thr308 (*Figure 5D*). In accordance with these observations, phosphorylation of mTORC1 substrate, 4E-BP1 was normal in *Napa*$^{hyh/hyh}$ CD4 T cells (*Figure 5E*). Of note, Orai1 depletion restored AKT Ser473 phosphorylation in *Napa*$^{hyh/hyh}$ CD4 T cells (*Figure 5F*) and the levels of mTORC2 complex proteins were largely comparable in WT and *Napa*$^{hyh/hyh}$ CD4 T cell WCLs (*Figure 5G*). These data demonstrate that Orai1 mediated sodium influx and the consequent drop in [ATP]$_i$ disrupts TCR-induced mTORC2 activation.

## mTORC2 regulates NFκB activation *via* multiple signaling intermediates in *Napa*$^{hyh/hyh}$ CD4 T cells

mTORC2 has been shown to regulate NFκB activation *via* many different signaling intermediates including AGC kinase, AKT (*Masui et al., 2014*), PKC-θ and IκB-α (*Lee et al., 2010*; *Tanaka et al., 2011*). Indeed, in addition to AKT Ser473 (*Figure 5A*), phosphorylation of PKC-θ (*Figure 6A*), IKK-β (*Figure 6B*) and IκB-α (*Figure 6C*) was also reduced in TCR-stimulated *Napa*$^{hyh/hyh}$ CD4 T cells. Of note, similar to AKT Ser473 (*Figure 5F*), IκB-α phosphorylation was restored in *Napa*$^{hyh/hyh}$ CD4 T cells upon ablation of Orai1 (*Figure 6D*). Given that monensin can also inhibit NFκB activation (*Deng et al., 2015*), our data demonstrate that defective mTORC2 signaling results in the inhibition of nuclear translocation of NFκB in *Napa*$^{hyh/hyh}$ CD4 T cells.

c-rel-/- mice harbor a significant defect in the development as well as function of Foxp3 T cells (*Isomura et al., 2009*; *Ruan et al., 2009*; *Long et al., 2009*). Paradoxically, ablation of mTORC2 complex proteins enhances Foxp3 T cell development (*Delgoffe et al., 2009*) (*Lee et al., 2010*). Indeed, despite a strong defect in c-Rel and NFκB p65 activation, we observed only a partial decrease in *Napa*$^{hyh/hyh}$ Foxp3 T cell development in vivo and well as in vitro (*Figure 2*). To resolve

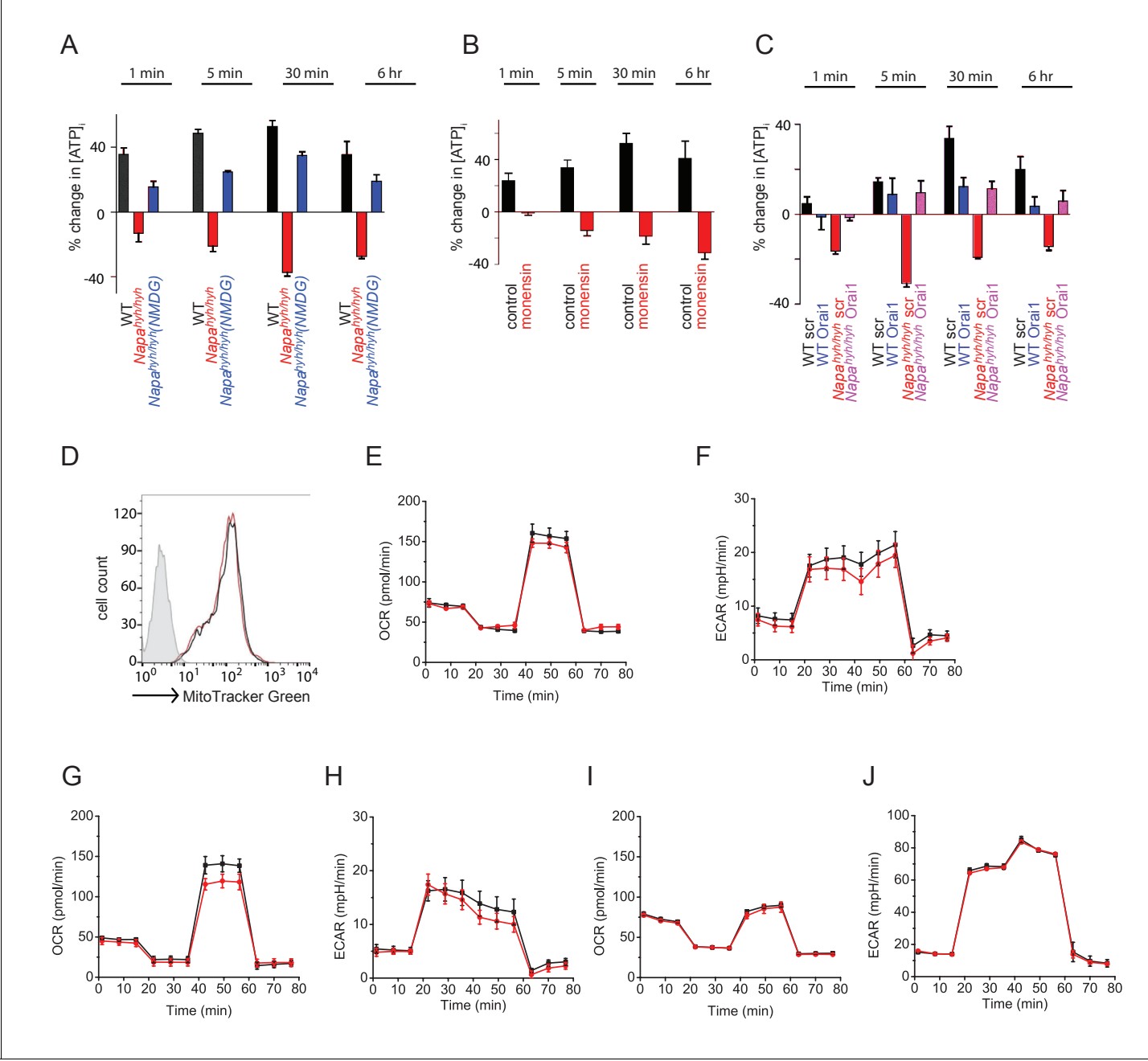

**Figure 4.** TCR-induced non-specific sodium influx depletes [ATP]$_i$ in *Napa*$^{hyh/hyh}$ CD4 T cells. (**A**) Percent change in intracellular ATP levels [ATP]$_i$ in anti-CD3-stimulated WT and *Napa*$^{hyh/hyh}$ CD4 T cells, measured at different times post-stimulation. (n = 6). (**B**) Percent change in [ATP]$_i$ in WT CD4 T cells, stimulated with anti-CD3 in the presence or absence of Monensin. (n = 2). (**C**) Percent change in [ATP]$_i$ in anti-CD3-stimulated WT and *Napa*$^{hyh/hyh}$ CD4 T cells, treated with scramble (scr) or Orai1 RNAi (Orai1). (n = 2). (**D**) FACS profiles of WT (black) and *Napa*$^{hyh/hyh}$ (red) CD4 T cells stained with Mitotracker green. (n = 2). (**E–J**) OCR and ECAR profiles of naive (**E,F**), TCR receptor stimulated for 6 hr (**G,H**), or TH0 (**I,J**) WT (black) and *Napa*$^{hyh/hyh}$ (red) CD4 T cells. (n = 2 each).

this conundrum, we explored other known targets of mTORC2 and found that the nuclear export of FOXO-1 was also partially inhibited (*Figure 6E*). FOXO-1 is necessary for Foxp3 Treg development (*Kerdiles et al., 2010*). However, its inactivation was recently shown to be required for the activation and tumor infiltration of Foxp3 Tregs in vivo (*Luo et al., 2016*). In agreement with those findings,

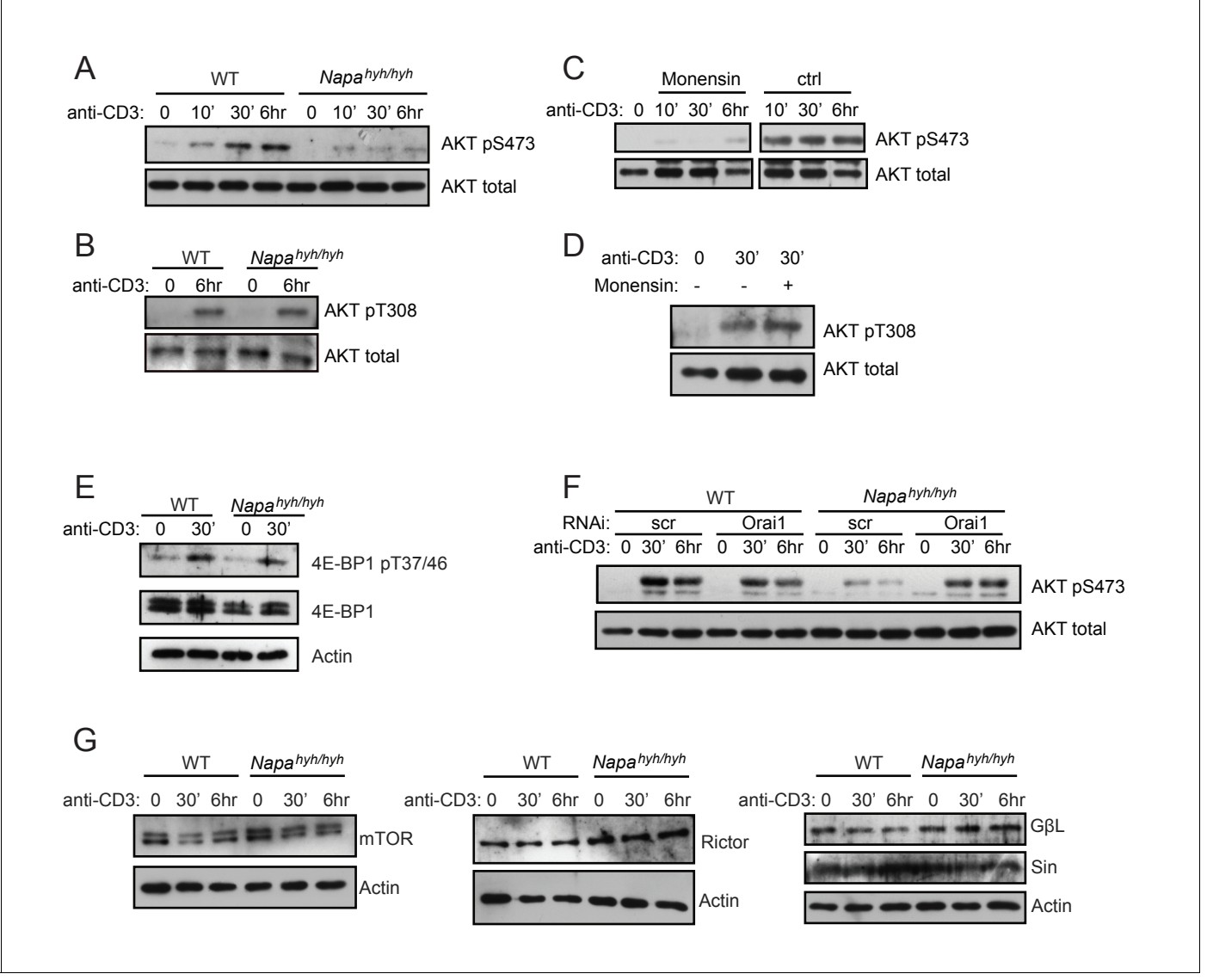

**Figure 5.** Depletion of [ATP]ᵢ inhibits mTORC2 activation in *Napa*<sup>hyh/hyh</sup> CD4 T cells. (A,B) Western blot for total and pS473 (**A**) or pT308 (**B**) phospho-AKT in receptor-stimulated WT and *Napa*<sup>hyh/hyh</sup> CD4 T cell WCLs at different times post-activation. (n = 3). (C,D) Western blot for total and pS473 phospho-AKT (**C**) or pT308 phospho-AKT (**D**) in WT CD4 T cell receptor stimulated in the presence or absence of monensin. (n = 2). (**E**) Western blot for total and phospho- pT37/46 4E-BP1 in receptor-stimulated WT and *Napa*<sup>hyh/hyh</sup> CD4 T cell WCLs. (**F**) Western blot for total and pS473 phospho-AKT in WCLs of receptor-stimulated WT and *Napa*<sup>hyh/hyh</sup> CD4 T cells, treated with scramble (scr) or Orai1 RNAi (Orai1). (n = 2). (**G**) Western blot for mTORC2 complex proteins in the WCLs of receptor-stimulated WT and *Napa*<sup>hyh/hyh</sup> CD4 T cells. (n = 2).

reduced export of FOXO-1 may partially restore Foxp3 expression in *Napa*<sup>hyh/hyh</sup> iTregs, but inhibit their activation and expansion in vivo.

## TCR-stimulated *Napa*<sup>hyh/hyh</sup> CD4 T cells show significantly altered gene expression

To analyze the cumulative effect of a concomitant defect in the activation of NFAT, NFκB and nuclear export of FOXO-1 on gene expression, we performed RNA sequencing on TCR-stimulated wildtype and *Napa*<sup>hyh/hyh</sup> CD4 T cells (*Figure 6F and G*). Principal component analysis (PCA) on *Napa*<sup>hyh/hyh</sup> and WT samples showed that gene expression from *Napa*<sup>hyh/hyh</sup> replicates were highly correlated between themselves and clustered distinctly from wildtype samples (*Figure 6F*). ~500 genes from TCR receptor-stimulated *Napa*<sup>hyh/hyh</sup> CD4 T cells were up or downregulated by >2 fold

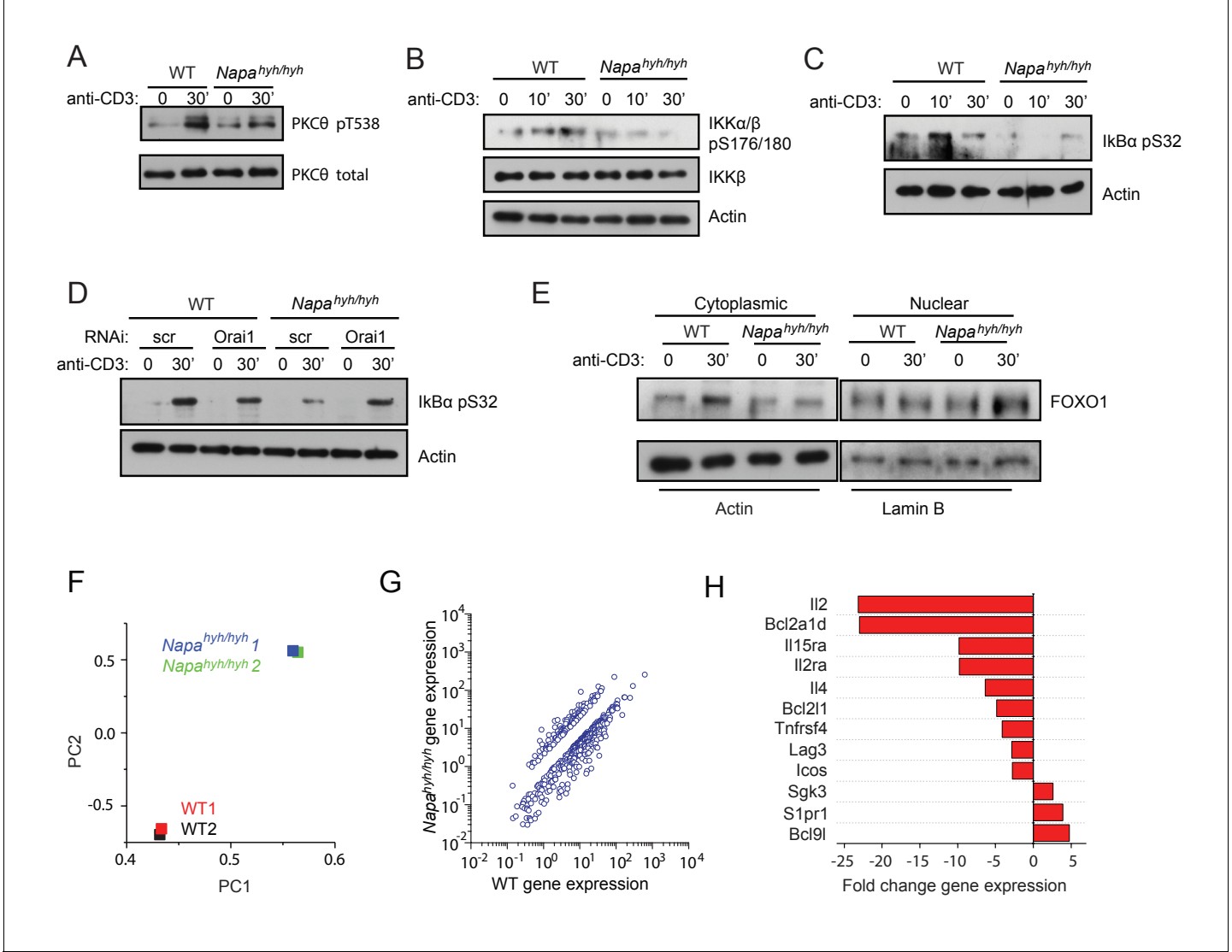

**Figure 6.** mTORC2 regulates NFκB activation *via* multiple signaling intermediates in *Napa^hyh/hyh* CD4 T cells. (A–C) Western blots for total and pT538 phospho-PKC-θ (A), phospho-IKKβ (B) and phospho-IκB-α (C) in WCLs of receptor-stimulated WT and *Napa^hyh/hyh* CD4 T cells. (n = 2). (D) Western blot for total and phospho-IκB-α in WCLs of receptor-stimulated WT and *Napa^hyh/hyh* CD4 T cells, treated with scr or Orai1 RNAi. (E) Western blot for cytosolic and nuclear FOXO-1 in receptor-stimulated WT and *Napa^hyh/hyh* CD4 T cells. (n = 3). (F) Principal component analysis (PCA) on gene expression data from TCR-stimulated WT and *Napa^hyh/hyh* CD4 T cell RNA. (n = 2 from 2 independent chimeras each). (G) Scatter plot showing the normalized means of gene expression values (RPKM) in *Napa^hyh/hyh* and WT CD4 T cells after filtering for genes as described in Materials and methods. (H) Bar plot showing fold change in the expression of a few representative genes between WT and *Napa^hyh/hyh* samples from (F,G) (See also *Figure 6—source data 1*).

The following source data is available for figure 6:

**Source data 1.** Pathways defective in receptor stimulated *Napa^hyh/hyh* CD4 T cells.

compared to their expression in WT cells. The list of differentially expressed genes can be found at (10.5061/dryad.202fn).

A scatter plot of these gene expression values from *Napa^hyh/hyh* and WT samples is shown in *Figure 6G* and the average fold change in the expression of a few representative targets of NFκB, NFAT and FOXO-1 are shown in *Figure 6H*. We grouped the differentially expressed genes into pathways using a pathway analysis software. The non-redundant pathways with at least 10% representation of total genes were considered significantly disrupted and top 50 of those are listed in

(*Figure 6—source data 1*). Thus, receptor-induced non-specific sodium influx disrupts a novel [ATP]$_i$→ mTORC2 signaling node in *Napa*$^{hyh/hyh}$ CD4 T cells, contributing to wide-spread and severe defects in CD4 T cell gene expression, effector cytokine production and Foxp3 regulatory T cell development. To our knowledge, an early rise in [ATP]$_i$ levels upon TCR stimulation, its sensitivity to sodium permeation and its direct role in the activation of mTORC2 signaling node have not been reported previously.

## Ectopic expression of α-SNAP can restore defects in *Napa*$^{hyh/hyh}$ CD4 T cell effector cytokine production

Previous characterization of developmental defects in the neuroepithelium of *Napa*$^{hyh/hyh}$ mice has suggested that *hyh* mutation causes reduced expression of α-SNAP due to mRNA instability, but M105I α-SNAP has normal function (*Chae et al., 2004*). Indeed, similar to *Napa*$^{hyh/hyh}$ CD4 T cells, RNAi-mediated reduction of α-SNAP expression inhibited SOCE (*Figure 7A*) and the expression of key effector cytokines (*Figure 7B*) in primary CD4 T cells. Yet, some recent studies have reported that purified M105I α-SNAP protein displays altered function in vitro (*Rodríguez et al., 2011*; *Park et al., 2014*). Therefore, we wondered whether protein intrinsic functional defects in M105I α-SNAP contribute to *Napa*$^{hyh/hyh}$ immunodeficiency. To test this, we reconstituted *Napa*$^{hyh/hyh}$ CD4 T cells with WT or M105I α-SNAP. Both WT as well as M105I mutant α-SNAP fully restored intracellular IL-2 production (*Figure 7C*) as well as SOCE (*Figure 7D*) in *Napa*$^{hyh/hyh}$ T cells. Consistent with these data, purified M105I α-SNAP protein bound to Stim1 and Orai1 as efficiently as WT α-SNAP in pull down assays in vitro (*Figure 7E*). Further, ectopic expression of M105I α-SNAP in HEK 293 cells revealed its cytosolic localization in resting cells (*Figure 7F*) as well as co-clustering with Stim1 in ER-PM junctions of store-depleted cells (*Figure 7G*), identical to WT α-SNAP localization patterns observed previously (*Miao et al., 2013*). Taken together, these data show that M105I α-SNAP is functionally similar to WT α-SNAP in its ability to support SOCE and CD4 T cell gene expression.

*Figure 8* summarizes the signaling nodes affected by TCR-induced non-specific sodium influx in α-SNAP deficient, *Napa*$^{hyh/hyh}$ CD4 T cells contributing to severely altered gene expression, reduced production of CD4 T cell effector cytokines and Foxp3 Treg development.

## Discussion

We have shown that TCR-induced, Orai1-mediated sodium influx disrupts a novel ATP- dependent signaling cascade necessary for the development of Foxp3 regulatory T cells. High extracellular sodium has been previously shown to upregulate T helper 17 differentiation (*Wu et al., 2013*; *Kleinewietfeld et al., 2013*). However, to our knowledge, signaling and phenotypic defects resulting from TCR-induced non-specific sodium influx *via* a well-characterized calcium channel have not been explored previously. Given that deletion or functional ablation of Orai1 inhibits a linear signaling pathway culminating in NFAT activation (*Feske et al., 2006*), *Napa*$^{hyh/hyh}$ mice would be an excellent model for further analyses of in vivo phenotypes resulting from permeation and ion selectivity defects in CRAC channels of mice and humans.

Our findings may also provide mechanistic insights into the previous association of elevated expression of α-SNAP with some aggressive forms of colorectal cancer (*Grabowski et al., 2002*). Likewise, monensin-mediated inhibition of Foxp3 iTreg development could, in part, explain the mechanisms underlying its effective re-purposing in the treatment of several different types of cancers (*Deng et al., 2015*) (*Tumova et al., 2014*).

The Na K ATPase is ubiquitously expressed and during periods of heightened cellular activity, such as action potentials in neurons, it is estimated to consume >70% of [ATP]$_i$. ATP hydrolysis is therefore used as a reliable readout for the Na K ATPase activity (*Weigand et al., 2012*). Indeed, sodium influx in TCR-stimulated *Napa*$^{hyh/hyh}$ CD4 T cells correlated well with reduced [ATP]$_i$ levels in our study and no additional defects were observed in the mitochondrial content or morphology (*Li et al., 2016*). Therefore, it is reasonable to conclude that depletion of [ATP]$_i$ resulted from increased Na K ATPase activity in receptor-stimulated *Napa*$^{hyh/hyh}$ CD4 T cells. Because Orai1 ablation prevented sodium influx, [ATP]$_i$ depletion and reversed mTORC2 signaling defects in *Napa*$^{hyh/hyh}$ CD4 T cells, these data conclusively demonstrate that sodium permeation *via* Orai1 depletes [ATP]$_i$.

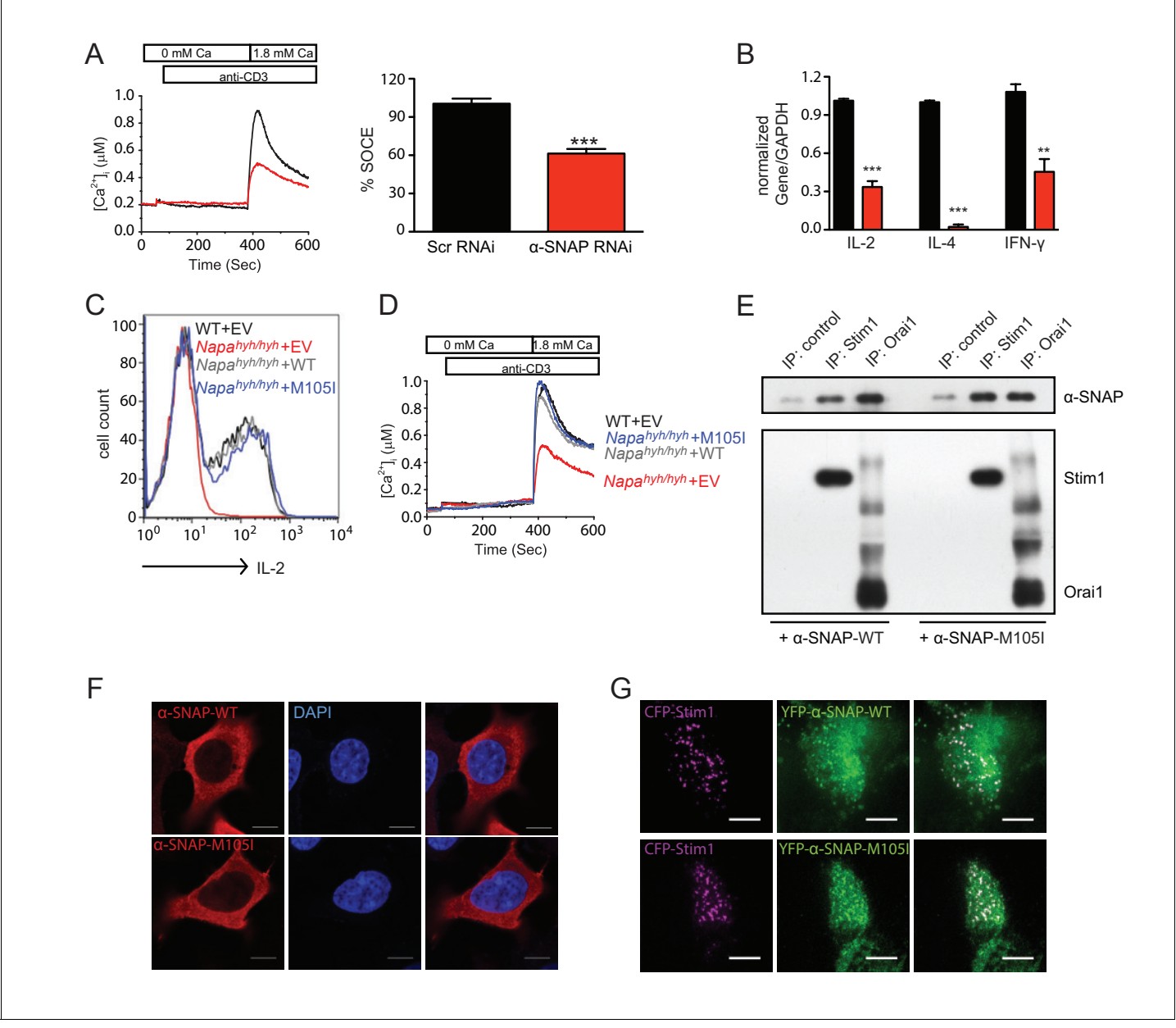

**Figure 7.** Ectopic expression of α-SNAP can restore defects in *Napa^hyh/hyh* CD4 T cells. (**A**) Average cytosolic calcium levels, measured using FURA 2AM, in scr (black) and α-SNAP RNAi (red)-treated cells stimulated with anti-CD3 antibody to measure SOCE. (n = 3 with ~50 to 100 cells per experiment). (**B**) Quantitative PCR to estimate the expression of key effector cytokines in scr (black) and α-SNAP RNAi (red)-treated Th0 cells. (n = 2 repeats; samples from 3 repeats of RNAi). (**C**) Representative FACS profiles showing intracellular IL-2 staining in WT and *Napa^hyh/hyh* CD4 T cells reconstituted with EV, WT or M105I α-SNAP. (n = 3). (**D**) Average cytosolic calcium levels, measured using FURA 2AM, in anti-CD3- stimulated WT and *Napa^hyh/hyh* CD4 T cells expressing empty vector (EV), WT or M105I α-SNAP. (n = 2 with ~50 to 200 cells each). (**E**) Western blot showing in vitro binding of WT and M105I α-SNAP to Stim1 and Orai1. (n = 2). (**F**) Confocal images of HEK293 cells expressing WT or M105I α-SNAP and stained with anti-α-SNAP antibody and DAPI (Scale bar 10 μm). (n = 2; 5 to 6 cells/ per group/ experiment). (**G**) TIRF images of store-depleted HEK 293 cells co-expressing CFP-Stim1 and YFP-tagged WT or M105I α-SNAP. (Scale bar 10 μm). (n = 2 with 5 to 6 cells/ per group/ experiment).

Following antigen receptor stimulation, surplus [ATP]$_i$ has been shown to get exported from T cells and bind P2X receptors to sustain calcium influx in an autocrine manner (*Schenk et al., 2008*; *Yip et al., 2009*). Thus, a decrease in [ATP]$_i$ could further compound the defect in sustained calcium flux and NFAT activation in *Napa^hyh/hyh* CD4 T cells.

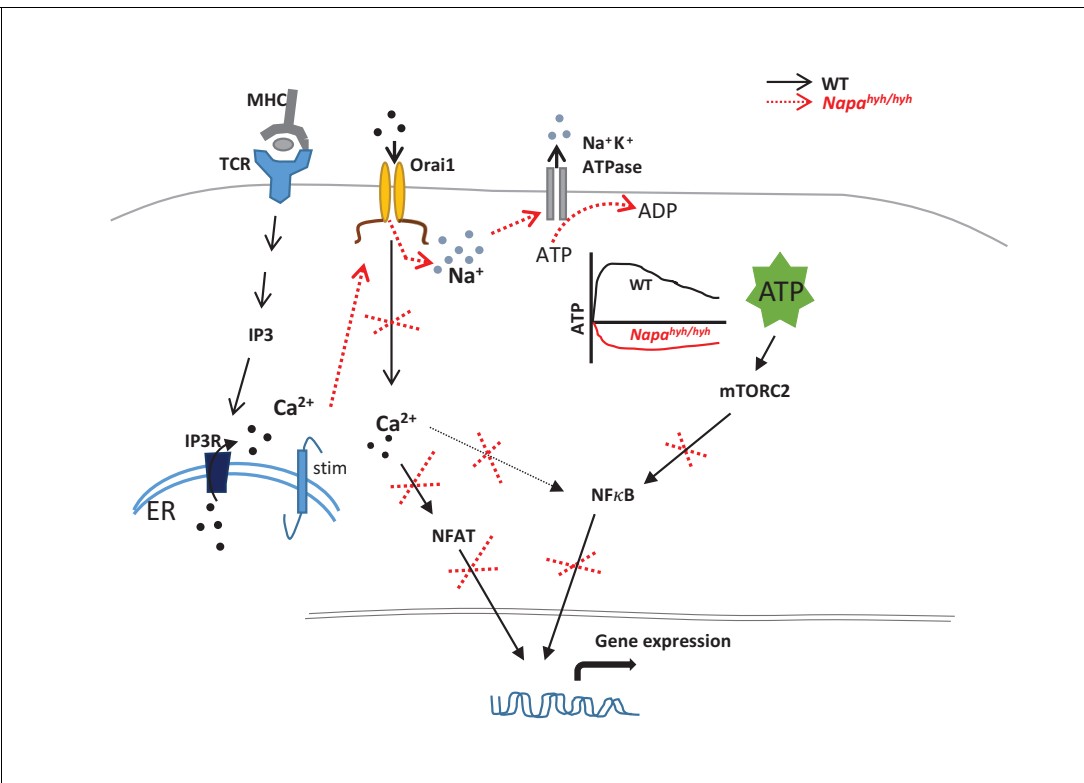

**Figure 8.** Summary of signaling nodes affected by TCR induced non-specific sodium influx.

Genetic ablation of individual components of mTORC2 complex has demonstrated its important role in CD4 T cell homeostasis as well as helper T cell and Foxp3 Treg differentiation (*Gamper and Powell, 2012*; *Chapman and Chi, 2014*; *Masui et al., 2014*; *Delgoffe et al., 2009*; *Navarro and Cantrell, 2014*). However, because the upstream activator of mTORC2 was unestablished in T cells (*Navarro and Cantrell, 2014*; *Masui et al., 2014*), its role could be more complex and context-dependent in vivo. Adding to this complexity, mTORC2 regulates a variety of downstream targets (*Laplante and Sabatini, 2009*, *2012*). For instance, mTORC2→NF-κB signaling is involved in cancer progression downstream of EGFR (*Tanaka et al., 2011*). Intriguingly, while mTORC2 inhibits Foxp3 Treg differentiation (*Delgoffe et al., 2009*), NFκB has been shown to be necessary for Treg development and function (*Isomura et al., 2009*; *Ruan et al., 2009*; *Long et al., 2009*). Likewise, FOXO-1 has a dual role in Treg development versus activation (*Kerdiles et al., 2010*; *Luo et al., 2016*). While its ablation inhibits Foxp3 Treg development, its inactivation is necessary for Treg activation, homing and tumor infiltration (*Kerdiles et al., 2010*; *Luo et al., 2016*). Therefore, future analyses of *Napa^{hyh/hyh}* mice will help in evaluating the consequences of simultaneous inhibition of NFAT and mTORC2-dependent signaling pathways on CD4 T cell homeostasis, differentiation and function in specific physiological and disease contexts in vivo.

It is intriguing that membrane trafficking of TCR and co-receptors was normal in *Napa^{hyh/hyh}* CD4 T cells. This result can be explained by considering a competitive binding model for the interactions of α-SNAP with membrane trafficking proteins *versus* the Orai-Stim complex. Indeed, a significant fraction of α-SNAP has been shown to be sequestered by its binding to monomeric syntaxins in resting cells (*Bátiz et al., 2009*; *Rodríguez et al., 2011*). Furthermore, we have found that the affinity of α-SNAP for SNAREs is significantly higher when compared to Stim1 and Orai1 (Bhojappa et al. unpublished findings). Higher affinity and constitutive association of α-SNAP with SNAREs could explain the relatively intact membrane trafficking of proteins in *Napa^{hyh/hyh}* CD4 T cells which harbor only a partial depletion of α-SNAP levels when compared to more robust defects in supporting CRAC channel function. (*Bronson and Lane, 1990*; *Chae et al., 2004*; *Hong et al., 2004*).

In summary, we have identified a novel [ATP]$_i\rightarrow$ mTORC2-dependent signaling axis, demonstrated its requirement for CD4 and Foxp3 regulatory T cell differentiation and established its sensitivity to non-specific sodium influx *via* Orai1.

## Materials and methods

### Mice

*Napa*$^{hyh/hyh}$ (RRID: MGI:3033683) mice were obtained from Jackson Laboratory (Bar Harbor, ME) (strain number: 001035, strain name: B6C3Fe *a/a-Napa*$^{hyh/hyh}$/J) and backcrossed onto C57BL/6J until they were >99.4% C57BL/6J, with the help of speed congenics core (RDCC) of the Washington University. All animal experiments were performed according to the guidelines of the Animal Studies Committee of the Washington University School of Medicine in Saint Louis, Protocol Approval Number 20150289.

### Genotyping

For genotyping *Napa*$^{hyh/hyh}$ mice a Custom TaqMan SNP Genotyping Assay developed by Applied Biosystems (Foster City, CA) was used. Forward primer: TCTTTGCTCCCTAGAGGCCATTA, Reverse primer: CAAGCAACCCTTACCATGTCTGTAT, Reporter 1 (VIC): CTGTCTGATGAGAGCAA, Reporter 2 (FAM): ACTGTCTGATAAGAGCAA.

### Fetal liver chimeras

CD45.1 male mice from Charles River were used as recipients for fetal liver chimeras. Recipient mice were irradiated at 850 rads and fetal livers extracted from E15.5 wildtype or *Napa*$^{hyh/hyh}$ donor embryos were injected into 3 to 4 recipients each. At least 3 to 4 chimeras were analyzed per experiment and a total of 30 to 40 chimeras of each group were generated and analyzed as part of the entire study. All chimeras were analyzed 8 to 12 week post-reconstitution.

### Technical and biological replicates

Unless otherwise specified within figure legends, 'n' denotes technical as well as biological replicates. For instance, n = 3 means three technical repeats from three independent chimera pairs across 2 to 3 injections.

### Cell isolation from chimeras and CD4 T cell sorting

For each experiment, spleen and lymph nodes were harvested from wildtype and *Napa*$^{hyh/hyh}$ fetal liver chimeras and subjected to two step sorting. CD4+ T cells were first enriched using MACS CD4 + T cell purification kit (Miltenyi Biotec Inc., San Diego, CA) according to manufacturer's instructions. CD4 T cell purity was routinely >95%. To obtain unperturbed CD45.2+CD4+ double positive cells, MACS-enriched cells were stained with anti-CD45.1 APC and sorted using Aria II (BD Biosciences, San Jose, CA) by gating on CD45.1 negative cells.

### Measurement of single cell SOCE and [Ca$^{2+}$]$_i$

CD45.2+CD4+ T lymphocytes were sorted from chimeras and plated on coverslips. Cells were loaded with 1 μM Fura-2-AM (Life Technologies, Eugene, OR) in Ringer's buffer (135 mM NaCl, 5 mM KCl, 1 mM CaCl2, 1 mM MgCl2, 5.6 mM Glucose, and 10 mM Hepes, pH 7.4) for 40 min in the dark, washed, and used for imaging. Baseline images were acquired for 1 min and then cells were simulated with 10 μg/ml soluble anti-CD3 (Biolegend, San Diego, CA) plus 5 μg/ml secondary antibody (Biolegend) and imaged simultaneously in nominally calcium free Ringer's buffer for 5 to 6 min. Subsequently, extracellular calcium was replenished, and cells were imaged for an additional 5–6 min. 50 to 200 cells were analyzed per group in each experiment. An Olympus IX-71 inverted microscope equipped with a Lamda-LS illuminator (Sutter Instrument, Novato, CA), Fura-2 (340/380) filter set (Chroma, Bellows Falls, VT), a 10 × 0.3 NA objective lens (Olympus, UPLFLN, Japan), and a Photometrics Coolsnap HQ2 CCD camera was used to capture images at a frequency of ~1 image pair every 1.2 seconds. Data were acquired and analyzed using MetaFluor (Molecular Devices, Sunnyvale, CA), Microsoft Excel, and Origin softwares. To calculate [Ca]$_i$, Fura-2 Calcium Imaging Calibration Kit (Life technologies) was used according to manufacturer's instructions. Briefly, standard samples

containing dilutions of free $Ca^{2+}$ (0 to 39 μM) were imaged as described above to obtain the constant $K_d$. $[Ca^{2+}]_i$ was then determined using the following equation:

$$[Ca^{2+}] = K_d \times \frac{[R - R_{min}]}{[R_{max} - R]} \times \frac{F^{380}_{max}}{F^{380}_{min}}$$

where R is the ratio of 510 nm emission intensity with excitation at 340 nm versus 380 nm; $R_{min}$ is the ratio at zero free $Ca^{2+}$; $R_{max}$ is the ratio at saturating free $Ca^{2+}$; $F^{380}_{max}$ is the fluorescence intensity with excitation at 380 nm, for zero free $Ca^{2+}$; and $F^{380}_{min}$ is the fluorescence intensity at saturating free $Ca^{2+}$. SOCE was calculated as (SOCE=highest $[Ca^{2+}]_i$ – basal $[Ca^{2+}]_i$), where highest $[Ca^{2+}]_i$ was the highest value after replenishing extracellular calcium and basal $[Ca^{2+}]_i$ was the lowest $[Ca^{2+}]_i$, following store-depletion in calcium-free buffer. Percentage of average SOCE in $Napa^{hyh/hyh}$ or α-SNAP RNAi-treated samples was then determined by setting the average of wildtype SOCE to 100%.

## Measurement of single cell [Na]$_i$

CD45.2+CD4+ T cells were sorted from chimeras, plated on coverslips and loaded with 2.5 μM SBFI-AM (Life Technologies) in Hank's balanced salt solution (HBSS) buffer at room temperature for 40 min in the dark, washed and used for imaging. Baseline images were acquired for 1 minute, and then cells were stimulated with 10 μg/ml soluble anti-CD3 (Biolegend) plus 5 μg/ml secondary antibody (Biolegend) and imaged simultaneously in HBSS buffer. SBFI was alternatively excited at 340 and 380 nm, and images were collected at 510 nm emission wavelength using the microscope setup described above. Nearly 150 cells were analyzed per group. To calculate $[Na^+]_i$, SBFI was calibrated in vivo in T lymphocytes based on the protocol described previously (*Negulescu and Machen, 1990*; *Donoso et al., 1992*). Briefly, cells were loaded with SBFI and imaged in the buffer containing serial dilutions of free $[Na^+]$ concentration ranging from 0 and 150 mM, which were obtained by mixing $Na^+$ free (130 mM potassium gluconate and 30 mM KCl) and $Na^+$ MAX (130 mM sodium gluconate and 30 mM NaCl) solutions. To equilibrate extracellular and intracellular sodium, cells were treated with monovalent cation ionophore gramicidin D at 5 μM. After imaging cells in at least five dilutions, standard curve was obtained by plotting $[Na^+]$ on (x-axis) versus $[Na^+]/(1/R_0-1/R)$ on (y-axis), where R is the ratio of emission intensity at 510 nm with excitation at 340 nm versus 380 nm; $R_0$ is the ratio at zero $Na^+$. From the above equation, the apparent $K_d$ of SBFI in T lymphocytes was obtained and $[Na^+]_i$ of experimental samples was then calculated using the constants derived from the regression line.

## Intracellular ATP quantification

CD45.2+CD4+ T lymphocytes were sorted from chimeras and stimulated with 10 μg/ml anti-CD3, 5 μg/ml secondary antibody and 2 μg/ml anti-CD28 for indicated times. Subsequently, cells were washed using cold HBSS, pelleted and boiled in 100 μl TE buffer at 95°C for 5 to 7 minutes, and spun at 14000 RPM for 3 minute. Supernatants containing intracellular ATP and ATP standard were diluted using ATP assay solution according to manufacturer's instructions (ATP Determination Kit, Molecular Probes, Eugene, OR). Luminescence in standard and experimental samples was measured using FlexStation III, and intracellular ATP in experimental samples was calculated using ATP standard curve.

## Whole cell lysates (WCLs), SDS-PAGE and Western blot

CD45.2+CD4+ T lymphocytes were sorted from chimeras and stimulated with 10 μg/ml soluble anti-CD3 plus 5 μg/ml secondary antibody, and 2 μg/ml soluble anti-CD28 (Biolegend) in HBSS at 37°C for indicated times. Post-stimulation, cells were suspended in cold HBSS, pelleted down and lysed in RIPA buffer (Cell signaling, Danvers, MA). WCLs were boiled with Laemmli sample buffer containing 100 mM DTT and resolved using 10 or 12% SDS-polyacrylamide gel. Proteins were transferred by Western blotting to nitrocellulose membrane and probed with respective antibodies as described previously (*Miao et al., 2013*). Antibodies used in this paper (*Source data 1*).

## Nuclear and cytosolic extracts

CD45.2+CD4+ T lymphocytes were sorted from chimeras and stimulated with 10 µg/ml soluble anti-CD3 (Biolegend) and 5 µg/ml secondary antibody along with 2 µg/ml soluble anti-CD28 (Biolegend) for 30 min. Cytoplasmic and nuclear extracts were prepared using (Thermo Scientific, Rockford, IL) NE-PER kit as per manufacturer's instructions and subjected to SDS-PAGE gel and Western blot as described previously (*Miao et al., 2013*).

## Gene expression analysis

Total RNA was extracted from cells by using RNeasy mini kit (QIAGEN, Germany), reverse transcribed to cDNA with M-MLV RT-PCR (Promega, Madison, WI) and used for Q-PCR. GAPDH and 18S rRNA were first used as housekeeping genes for normalization of expression. Because Ct values for GAPDH were closer to the Ct values of genes being analyzed here, final normalization was done using GAPDH.

## RNA sequencing

CD45.2+CD4+ T lymphocytes were sorted from chimeras using MACS beads and BD Aria II and stimulated using plate-coated 10 µg/ml anti-CD3 along with 2 µg/ml soluble anti-CD28 for 6 hr. Following stimulation, total RNA was extracted by using RNeasy mini kit (QIAGEN) and submitted for quantification, library preparation, sequencing, and initial bioinformatics analysis to Genewiz (South Plainfield, NJ). Briefly, RNA samples were quantified using Qubit 2.0 Fluorometer (Life Technologies, Carlsbad, CA) and RNA integrity was checked with 2100 Bioanalyzer (Agilent Technologies, Palo Alto, CA). Whole transcriptome RNA enrichment was performed using Ribozero rRNA Removal Kit (1:1 mixture of Human/Mouse/Rat probe and Bacteria probe) (Illumina, San Diego, CA). For RNA sequencing library preparation, NEB Next Ultra RNA Library Prep Kit for Illumina was used by following the manufacturer's recommendations (NEB, Ipswich, MA). Briefly, enriched RNAs were fragmented for 15 min at 94°C. First strand and second strand cDNA were subsequently synthesized. cDNA fragments were end repaired and adenylated at 3'ends, and universal adapter was ligated to cDNA fragments, followed by index addition and library enrichment with limited cycle PCR. Sequencing libraries were validated using a DNA Chip on the Agilent 2100 Bioanalyzer (Agilent Technologies), and quantified by using Qubit 2.0 Fluorometer (Invitrogen, Carlsbad, CA) as well as by quantitative PCR (Applied Biosystems, Carlsbad, CA).

The sequencing libraries were multiplexed, clustered on a single flow cell and loaded on the Illumina HiSeq 2500 instrument. Samples were sequenced using a $1 \times 100$ Single Read (SR) Rapid Run configuration. Image analysis and base calling were conducted using the HiSeq Control Software (HCS) on the HiSeq 2500 instrument. Raw sequence data (.bcl files) generated from Illumina HiSeq 2500 were converted into fastq files and de-multiplexed using Illumina bcl2fastq v1.8.4 program. One mismatch was allowed for index sequence identification. Gene expression analysis was performed using the CLC Genomics Workbench software, by trimming sequence reads to remove low-quality bases at ends, followed by mapping sequence reads to the mouse genome (Refseq) and calculating gene expression values as Reads Per Kilobase of transcript per Million mapped reads (RPKM). Gene expression data were further filtered to remove transcripts using the following criterion: (i) students t-test p value > 0.05 between biological replicates, (ii) <2 fold change, (iii) genes with <10 total exon reads in wildtype group. The list of differentially expressed, filtered genes was deposited at Datadryad (10.5061/dryad.202fn). The data were also subjected to pathway analysis using the Metacore software (Thomson Reuters, NY) and top 50 pathways with a p-value<0.05 are displayed in *Figure 6—source data 1*.

## Intracellular cytokine staining

Naïve or differentiated T cells were stimulated with PMA (20 ng/ml) and Ionomycin (1 µg/ml) and brefeldin A (Biolegend) for 5 to 6 hr and stained using anti-CD4 or anti-CD8 surface markers (Biolegend). Subsequently, cells were fixed and permeabilized and incubated with anti-IL-2, anti-IL-4 or anti-IFN-γ antibodies (Biolegend) and analyzed using FACS Calibur or LSR Fortessa analyzers and Flow Jo software (BD Biosciences).

## T cell proliferations

Naïve CD45.2+CD4+ T lymphocytes were sorted from chimeras and labeled with 10 µM CFSE, washed and stimulated with 5 µg/ml plate-coated anti-CD3 along with 2 µg/ml soluble anti-CD28 for 72 hr, stained with anti-CD4 antibody and analyzed using LSR Fortessa flowcytometer. In some experiments, unfractionated splenocytes were stimulated with soluble anti-CD3 and anti-CD28 for 48 hr, pulsed with 1 µCi $^3$H thymidine for additional 12–16 hr, harvested and counted.

## Th1/Th2 differentiation

CD4 T cells were purified from chimeras and stimulated with 5 µg/ml plate-coated anti-CD3 and 2 µg/ml soluble anti-CD28 in the presence of cytokines and neutralizing antibodies for 2 days. For Th1: 20 ng/ml IL-2, 20 ng/ml IL-12, 10 µg/ml anti-IL-4 and for Th2: 20 ng/ml IL-2, 50 ng/ml IL-4 was used. After 48 hrs cells were washed and cultured in the above cocktail of cytokines and antibodies for an additional three days. Cells were then stimulated with 20 ng/ml PMA, 1 µg/ml ionomycin and brefelin A (Biolegend) for 5–6 hr and stained for Th1/Th2 signature cytokines or transcription factors T-bet/Gata-3 as mentioned above.

## Foxp3 Treg differentiation and staining

CD45.2+CD4+ T lymphocytes were sorted from chimeras and differentiated with 10 µg/ml plate coated anti-CD3, 2 µg/ml soluble anti-CD28 and 10 ng/ml TGF-$\beta$ for five days and analyzed. Thymocytes, lymph node, splenocytes or in vitro differentiated Treg cells were stained with anti-CD4 and anti-CD25 surface markers, then fixed and permeabilized with Fix/Perm buffers (Biolegend) and stained with Alexa647-FoxP3 (Biolegend).

## RNAi in primary T lymphocytes

$\alpha$-SNAP targeting sequence, (CGCCAAAGACTACTTCTTCAA), was subcloned into MSCV-LTRmiR30-PIG retroviral vector (Openbiosystems, Lafayette, CO). Viral supernatants were made according to manufacturer's instruction. For infections, T cells were stimulated with anti-CD3 for 24 hr prior to infection and spun with the viral supernatant and polybrene (8 µg/ml) at high speed for 90 min. GFP positive cells were analyzed 3 day post infection.

## T cell transfections

Naïve T cells were transfected using Amaxa Mouse T Cell Nucleofector Kit (Lonza, Switzerland) according to manufacturer's instructions. Cells were analyzed ~16 hr post transfection.

## Cell lines and transfection

HEK293 cells were obtained from (ATCC:CRL-1573)(RRID: CVCL_0045), expanded and cultured in DMEM containing 10% FBS, L-glutamine, non-essential amino acids and sodium pyruvate. Cells were co-transfected with CFP-Stim1 and YFP-$\alpha$-SNAP or YFP-$\alpha$-SNAP M105I expressing vectors, using Lipofectamine 2000 (Life technologies, USA) and imaged using TIRF illumination as described previously (*Miao et al., 2013*). Cell line stocks were tested for mycoplasma contamination using Lonza Mycoalert (Lonza) every few years.

## In vitro binding and western blotting

myc-tagged Orai1 and Stim1 proteins were immunoprecipitated from HEK293 (RRID: CVCL_0045) cells and beads were incubated with purified recombinant $\alpha$-SNAP WT or M105I for 1 hr at 4°C. Post-incubation, beads were washed three times and protein complexes eluted by boiling in SDS containing sample buffer and subjected to SDS-PAGE and western blotting as described previously (*Miao et al., 2013*).

## Statistical analysis

Statistical significance represented as p value was calculated using unpaired student's *t*-test, unless otherwise specified. *p<0.05, **p<0.01, ***p<0.001.

## Acknowledgements

We thank Grzegorz B Gmyrek, Leah Owens and Cathrine Miner for technical assistance. Yinan wang for help with generation of fetal liver chimeras. Chyi Song Hsieh for advice on Foxp3 regulatory T cell analysis. This work was supported in part by, NIH-NIAID grant AI108636 and ACS-RSG 14-040-01-CSM.

## Additional information

### Funding

| Funder | Grant reference number | Author |
| --- | --- | --- |
| American Cancer Society | ACS-RSG 14-040-01-CSM | Monika Vig |
| National Institutes of Health | AI108636 | Monika Vig |

The funders had no role in study design, data collection and interpretation, or the decision to submit the work for publication.

### Author contributions

YM, Conceptualization, Data curation, Formal analysis, Validation, Visualization, Methodology, Writing—original draft, Project administration; JB, Data curation, Methodology; AD, Data curation, Formal analysis, Methodology; MV, Conceptualization, Supervision, Funding acquisition, Methodology, Writing—original draft, Project administration, Writing—review and editing

### Author ORCIDs

Yong Miao, http://orcid.org/0000-0003-2614-1445
Adish Dani, http://orcid.org/0000-0002-5491-7709
Monika Vig, http://orcid.org/0000-0002-4770-8853

### Ethics

Animal experimentation: All animal experiments were performed according to the guidelines of the Animal Studies Committee of the Washington University School of Medicine in Saint Louis, Protocol Approval Number 20150289.

## Additional files

### Supplementary files

• Source data 1. Antibodies used in this paper.

### Major datasets

The following dataset was generated:

| Author(s) | Year | Dataset title | Dataset URL | Database, license, and accessibility information |
| --- | --- | --- | --- | --- |
| Yong Miao, Jaya Bhushan, Adish Dani, Monika Vig | 2017 | Data from Na+ Influx via Orai1 Inhibits Intracellular ATP Induced mTORC2 Signaling To Disrupt CD4 T Cell Gene Expression and Differentiation | http://dx.doi.org/10.5061/dryad.202fn | Available at Dryad Digital Repository under a CC0 Public Domain Dedication |

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
