## [Decision Letter]

Thank you for submitting your article "T Cell Receptor Induced Na^+^ Influx Disrupts a Novel ATP → mTORC2 Signaling Node to Inhibit Regulatory T Cell Development" for consideration by *eLife*. Your article has been favorably evaluated by Tadatsugu Taniguchi (Senior Editor) and three reviewers, one of whom is a member of our Board of Reviewing Editors. The reviewers have opted to remain anonymous.

The reviewers have discussed the reviews with one another and the Reviewing Editor has drafted this decision to help you prepare a revised submission

Summary:

An α-SNAP hypomorphic mutation causing hydrocephalous with hopping gait (*hyh*) resulted in increased sodium influx in TCR-stimulated T cells, which induced ATP reduction and blocked AKT-mTORC2 signaling. This altered T-cell signaling impaired T-cell cytokine production and Treg cell development, showing a novel regulatory mechanism of Treg and T cell differentiation.

Essential revisions:

1) The authors claim that sodium influx affected not only Treg development but also Treg function. Defective Treg development was demonstrated, whereas the analysis of Treg function was not enough. The expression levels of Treg signature molecules, such as CTLA4, GITR and Helios, on *hyh* Treg cells should be assessed. As shown in Figure 2, the colitis model indicates *hyh* T cell-mediated inflammation was more severe than WT naïve T cell transfer. To evaluate the contribution of Treg, they should conduct co-transfer experiments with *hyh* or WT Tregs together with WT CD45RA-high CD4^+^ T cells, and assess Treg number and phenotype (expression of Treg functional molecules) of Foxp3+ cells. The authors also need to perform in vitro suppression assay to assess the function of *hyh* Treg cells.

2) Regarding the effects of the SNAP mutation on non-Treg cells, effector function of non-Treg cells should be tested in some experiments. Figure 1 indicates T-bet or GATA3 was higher in expression in *hyh* T cells. Thus, the authors should examine whether the development of in vitro iTreg is inhibited by these effector transcription factors. Additionally, *hyh* T cell transfer caused more severe pathology than WT cell transfer, although *hyh* T cells appeared to be defective in production of effector cytokines according to Figure 1. To address this discrepancy, the effector phenotype of T cells in vivo, such as cytokine production or T cell activation marker expression, should be evaluated in the colitis model.

3) This study proposes that increased sodium entry leads to reduced ATP levels in activated T cells that results in decreased MTORC2 activation. This would cause defective PKC-theta and NFkB activity that would lead to defective differentiation and function of Tregs. However, the data that shows that decreased MTORC2 activation due to reduced ATP prevents full NFkB activation is only correlative. To establish cause-effect, experiments would need to be designed to show the functional connection between all those observations. For example, would the defect in MTORC2 activation be bypassed by expressing a constitutively active PKC-theta? Or will the functional defects in those T cells be corrected by activating NFkB (for instance using an active form of IKK)? Nuclear translocation of p65 is independent of Orai1-mediated SOCE as suggested by Figure 3. If so, how about activations of upstream kinases or complex formation of adaptor molecules in *hyh* T cells? In addition, the authors should evaluate other NF-κB family proteins, especially c-rel. Also, can an ATP analog or an AKT activator (or constitutively active AKT) recover the reduction of in vitro iTreg development shown in Figure 2? To examine whether sodium influx reduces AKT activation by the depletion of intracellular ATP, it is necessary to show S473 phosphorylation in the presence of Digoxin or NMDG. How about silencing of Foxo1? With the demonstration that SNAP reconstitution improved the defects of IL-2 production in Figure 6, does it improve iTreg development?

4) The authors use monensin as the inducer of sodium influx, but monensin has many other biological functions such as inhibition of protein transport. For example, in Figure 5, they need to confirm that AKT pT308 is not affected by monensin, even though AKT pS473 is decreased. In addition, the effect of monensin on the iTreg development (Figure 3) is striking compared to the effect of SNAP mutation as shown in Figure 2, despite a similar degree of sodium influx and ATP reduction. The authors need to address whether the monensin-mediated inhibition can be rescued by AKT activator or ATP treatment.

5) While it is true that NFAT does not seem to be required for the generation of tTregs, the Vaeth et al. study showed an important decrease in the generation of pTregs in mice deficient in NFATc1 and NFTAc2. It might be difficult then to conclusively determine if the decreased generation of pTregs (or iTregs) is due to increased Na entry or decreased SOCE in *hyh* T cells.

---

## [Author Response]

*Essential revisions:*

*1) The authors claim that sodium influx affected not only Treg development but also Treg function. Defective Treg development was demonstrated, whereas the analysis of Treg function was not enough. The expression levels of Treg signature molecules, such as CTLA4, GITR and Helios, on hyh Treg cells should be assessed. As shown in Figure 2, the colitis model indicates hyh T cell-mediated inflammation was more severe than WT naïve T cell transfer. To evaluate the contribution of Treg, they should conduct co-transfer experiments with hyh or WT Tregs together with WT CD45RA-high CD4^+^ T cells, and assess Treg number and phenotype (expression of Treg functional molecules) of Foxp3+ cells. The authors also need to perform in vitro suppression assay to assess the function of hyh Treg cells.*

The colitis experiment was performed to evaluate CD4 T cell function in vivo and the phenotype we have observed can have several explanations, in addition to a defect in pTreg development and function. For instance, even though hyh CD4 T cell development was not obviously disrupted, repertoires as well as signaling thresholds could be altered as a result of several signaling and gene expression defects that we have found in *hyh* CD4 T cells. Future studies using *hyh* mice crossed to various TCR transgenic mice will allow us to differentiate between these possibilities in a concrete fashion.

We agree that Treg function was not directly assessed in this paper and have removed the colitis experiment from Figure 2 because extrapolation of the colitis model to Treg function is not a direct experiment to establish Treg function. We have modified the text and Discussion which now refers to Treg differentiation but not function. Future studies using adoptive co-transfers of WT and hyh Foxp3 Tregs using more classical models of assessment of Foxp3 Treg function will allow us to address the in vivo function of WT *vs hyh* Foxp3 Tregs.

Regarding surface expression of functional markers of Treg, we have found a partial but consistent defect in the surface expression of GITR in *hyh* Foxp3 Tregs. In addition, RNA sequencing of CD4 T cells revealed reduced expression of ICOS and LAG-3 in receptor activated *hyh* CD4 T cells. These data have been added to Figure 2 and Figure 6 respectively and suggest that there may be additional functional/ homing/ expansion defects in *hyh* Foxp 3 Tregs in vivo. Again, future studies will assess *hyh* Treg function.

*2) Regarding the effects of the SNAP mutation on non-Treg cells, effector function of non-Treg cells should be tested in some experiments. Figure 1 indicates T-bet or GATA3 was higher in expression in hyh T cells. Thus, the authors should examine whether the development of in vitro iTreg is inhibited by these effector transcription factors. Additionally, hyh T cell transfer caused more severe pathology than WT cell transfer, although hyh T cells appeared to be defective in production of effector cytokines according to Figure 1. To address this discrepancy, the effector phenotype of T cells in vivo, such as cytokine production or T cell activation marker expression, should be evaluated in the colitis model.*

T-bet and GATA-3 levels were not significantly different between experimental repeats. The average from three experiments is now shown in Figure 1—figure supplement 1. It is unclear why we do not see a decrease in GATA-3 levels despite a significant defect in mTORC2 signaling and IL-4 expression. One possibility is that another unknown factor/ binding partner prevents its turnover in *hyh* T cells.

Figure 1 shows that the defect in the expression of IFN-γ was only about 50% in *hyh* CD4 T cells. By contrast, the defect in IL-2 and the Th2 signature cytokine, IL-4, was much larger (See selected genes in Figure 6). Defective mTORC2 signaling could explain a relatively bigger defect in IL-4 expression in *hyh* CD4 T cells when compared to *Orai1-/-* mice. In this regard, our findings are in good agreement with previously published defects in *Orai1-/-* and Rictor-/- mice respectively (Vig, et al. Nat Imm. 2008; Gwack et al. MCB 2008; Lee et al. Immunity 2010; Delgoffe et al. Nat Immunol 2011).

Because IFN-γ is a major driver of colitis in the T cell adoptive transfer model, a 50% defect in its expression by *hyh* CD4 T cell effectors could easily have been compensated by several factors in vivo. For instance, the gut microenvironment is especially enriched in extracellular ATP, which has been shown to act as an adjuvant in vivo and drive inflammation (Atarashi et al. Nature, 2008; Borsellino et al. Blood 2007). Extracellular ATP could bind P2X receptors and induce calcium influx in *hyh* T cells in vivo, to bypass the defect in *hyh* CD4 T cell SOCE and IFN-γ production. It is therefore not unreasonable to speculate that depending on the environment, stimulus and levels of extracellular ATP, partial defects in SOCE can be compensated in vivo.

Future studies should be able to assess the in vivo effector function of *hyh* CD4 T cells in response to antigen specific stimulation after crossing *hyh* mice to various different TCR transgenic mice and also test their function in specific disease models.

*3) This study proposes that increased sodium entry leads to reduced ATP levels in activated T cells that results in decreased MTORC2 activation. This would cause defective PKC-theta and NFkB activity that would lead to defective differentiation and function of Tregs. However, the data that shows that decreased MTORC2 activation due to reduced ATP prevents full NFkB activation is only correlative. To establish cause-effect, experiments would need to be designed to show the functional connection between all those observations.*

MTORC2 mediated NFkB activation in CD4 T cells has been demonstrated by Lee et al. Immunity 2010, using Rictor-/- mice. Therefore, re-establishing cause-effect in the signaling pathway downstream of mTORC2 in CD4 T cells is going to be redundant with published literature.

Furthermore, downstream of mTORC2, NFkB can be activated by several independent routes i.e. via AKT, PKC-theta or directly via phosphorylated IKK. These routes of activation NFkB could be fully or partially redundant. Therefore, establishing cause-effect and functional connection using NFkB activators or by depleting individual upstream regulators of NFkB will affect several parallel and unrelated pathways.

Still, we have found that depleting Orai1 in *hyh* CD4 T cells reverses IκB-α phosphorylation. These data have been added to Figure 6.

*For example, would the defect in MTORC2 activation be bypassed by expressing a constitutively active PKC-theta? Or will the functional defects in those T cells be corrected by activating NFkB (for instance using an active form of IKK)?*

Please see our response above as well as below. First, the experiments suggested here have already been done by Lee at al. Immunity 2010. A major finding of our paper is the identification of [ATP]_i_ as the upstream regulator of mTORC2 in CD4 T cells and its sensitivity to non-specific Na influx via Orai1 and this has been established conclusively in the paper.

*Nuclear translocation of p65 is independent of Orai1-mediated SOCE as suggested by Figure 3. If so, how about activations of upstream kinases or complex formation of adaptor molecules in hyh T cells?*

We have found that in addition to PKC-theta, phosphorylation of IKK-β and IκB-α was also defective in *hyh* CD4 T cells. These data have now been added to Figure 6 and updated in the model in Figure 8. We have also found that depleting Orai1 reverses IKB-α phosphorylation (Figure 6). However, given the potential redundancy among upstream kinases that activate NFkB, establishing a cause-effect connection can be quite complex. For instance, previous studies have shown that mTORC2 can activate PKC-theta in CD4 T cells (Lee et al. 2010). However, NFκB has also been shown to be directly activated downstream of mTORC2 in Gliobalstoma cells (Tanaka et al. Cancer Discov. 2011).

*In addition, the authors should evaluate other NF-κB family proteins, especially c-rel.*

c-rel phosphorylation was defective in *hyh* CD4 T cells. The data have been added to Figure 3.

*Also, can an ATP analog or an AKT activator (or constitutively active AKT) recover the reduction of in vitro iTreg development shown in Figure 2?*

We are unaware of any cell permeable ATP analogs. In our hands, extracellular ATP cannot cross plasma membrane to compensate for the defects in ATP levels in *hyh* CD4 T cells (data not shown).

AKT is a repressor of thymic as well as iTreg differentiation (Haxinasto, S JEM, 2008). Therefore, it is unlikely that an AKT activator will compensate for the defects in iTreg development. We therefore think that reduced activation of AKT pS473 could be partially reversing or compensating for the severe defects in NFkB activation and Treg differentiation by stabilization of FOXO-1 in the nucleus.

*To examine whether sodium influx reduces AKT activation by the depletion of intracellular ATP, it is necessary to show S473 phosphorylation in the presence of Digoxin or NMDG.*

Digoxin not only inhibits Na K ATPase but has also been shown to directly binds ROR-γ T and influence TH17 differentiation (Huh, JR and Littman, D Nature 2011). The suggested experiment would therefore be difficult to interpret.

NMDG non-specifically blocks all types Na influx even in resting cells. This would block Na influx even via leak channels, which is detrimental for cells beyond short periods of time as it may disrupt the activity of many other ion channels. For this reason, we decided to deplete Orai1 expression in *hyh* CD4 T cells. Because Orai1 ablation largely blocked non-specific Na influx in TCR *hyh* T cells, it is a better, specific and much more conclusive experiment.

*How about silencing of Foxo1?*

FOXO-1 plays many different roles in T cells. It has been recently shown that graded activation of FOXO-1 is required for Foxp3 Treg activation (Luo et al. Nature 2016). FOXO-1 has also been reported to have a cytosolic role in hematopoietic differentiation (Wang et al. Nature Comm. 2016). It is currently unclear whether pharmacological inhibition of FOXO-1 acts in the same fashion as its deletion (Laine et a. JI, 2015; Kitz, A Embo 2016). Therefore, inhibiting or silencing FOXO-1 completely in *hyh* CD4 T cells is likely to yield ambiguous results. More so, because, we have only observed a partial defect in nuclear export of FOXO-1, it is therefore unclear how silencing it in *hyh* CD4 T cells will strengthen the main point of the paper. Perhaps for these reasons, although we do see upregulation of a few FOXO-1 target genes in Figure 6, others such as CTLA4 did not show up in our RNA sequencing analysis.

Because the Treg differentiation defect can be easily explained by a robust defect in NFkB, we have removed the AKT/FOXO-1 axis from our model in Figure 8, as this was not clearly established in our paper. Future studies may address its role, if any, in *hyh* CD4 T cells.

*With the demonstration that SNAP reconstitution improved the defects of IL-2 production in Figure 6, does it improve iTreg development?*

Unlike, IL-2 production (6 hour assay), iTreg differentiation requires 3-4 days of culture and stable expression. Therefore, we used retroviral reconstitution of α-SNAP expression in *hyh* CD4 T cells to address this question. However, due to technical issues which are likely due to the iTreg differentiation protocol, SNAP reconstitution efficiency was <5% in *hyh* CD4 T cells or if we modified the protocol in any way, we were unable to differentiate Foxp3 iTregs even in WT CD4 T cells. We have not been able to troubleshoot these technical issues.

*4) The authors use monensin as the inducer of sodium influx, but monensin has many other biological functions such as inhibition of protein transport. For example, in Figure 5, they need to confirm that AKT pT308 is not affected by monensin, even though AKT pS473 is decreased. In addition, the effect of monensin on the iTreg development (Figure 3) is striking compared to the effect of SNAP mutation as shown in Figure 2, despite a similar degree of sodium influx and ATP reduction. The authors need to address whether the monensin-mediated inhibition can be rescued by AKT activator or ATP treatment.*

Monensin treatment did not affect AKT pT308. These data have been added to the same figure (now Figure 6). We agree that monensin has several other effects besides increasing intracellular Na levels. Therefore, we have now shown a dose response in the inhibition of differentiation of iTreg with monensin (Figure 3).

We are unaware of any cell permeable ATP analogs. In our hands, extracellular ATP cannot cross the plasma membrane to compensate for depletion of intracellular ATP (data not shown).

AKT is a repressor of thymic as well as iTreg differentiation (Haxinasto, S JEM, 2008). Therefore, it is unlikely that AKT activator will compensate for the defects in iTreg development in monensin treated cells. Furthermore, monensin also has several other effects in addition to inducing Na influx. Monensin was used here just to support our findings and conclusions in *hyh* CD4 T cells.

5) While it is true that NFAT does not seem to be required for the generation of tTregs, the Vaeth et al. study showed an important decrease in the generation of pTregs in mice deficient in NFATc1 and NFTAc2. It might be difficult then to conclusively determine if the decreased generation of pTregs (or iTregs) is due to increased Na entry or decreased SOCE in hyh T cells.

We did not find any reduction in iTreg generation in Orai1 depleted CD4 T cells that also show a significant decrease in SOCE and NFAT translocation (Figure 3). Therefore, it is reasonable to conclude that a partial defect in SOCE alone does not reduce iTreg generation. Therefore, in *hyh* CD4 T cells, Foxp3 Treg differentiation defect largely results from increased sodium entry and the downstream defects in NFkB signaling.